# Not All Code Helps: Disentangling the Impact of Code Data on Mathematical Reasoning in Large Language Models

## Abstract

Incorporating code into training corpora has become a widely acknowledged practice in the development of modern foundation language models (LMs). Compared with a general Internet corpus, code offers high-quality, well-structured signals that substantially augment the coding proficiency of models. Beyond programming skills, prior research has suggested that code data may also contribute to non-coding capabilities. Nevertheless, through a series of rigorous controlled experiments, we demonstrate that the influence of code on other domains—particularly reasoning—remains limited. Our principal findings are as follows: (1) Code corpus yields substantial gains in programming-related abilities but only marginal improvements in non-coding tasks. We further observe that code competed with knowledge-intensive tasks. (2) Not all code data enhances the mathematical reasoning ability. We identify core subset that functions as cognitive scaffolding for mathematical reasoning, especially for complex problem-solving scenarios. (3) Formal reasoning (e.g., code reasoning or program-of-thought approaches) provides more pronounced improvements in challenging mathematical reasoning tasks, while natural language–based reasoning proves more effective for simpler reasoning problems. Finally, by probing the internal mechanisms of LMs, we reveal how training data modulates routing patterns, thereby shaping emergent model behavior. As a central driver of model capability, our findings disentangle domain-specific data into finer-grained, cross-domain ability dimensions and underscore promising directions for future data optimization.

## 1 Introduction

In general-purpose large language models (LLMs), code corpus typically constitutes 10% to 30% of the training corpus, rendering it a pivotal component of modern pretraining pipelines. Since the success of CodeX (Chen et al., 2021), which fine-tuned language models on code, followed by InstructGPT (Ouyang et al., 2022), which demonstrated that incorporating a modest proportion of code data into the training corpus can substantially enhance general-purpose LLMs, code has become a standard element in model development. With the rapid evolution of LLMs, training on code has consistently yielded improvements in capabilities such as Agents (Nakano et al., 2021; Kosinski, 2023; Chen et al., 2025), code reasoning (Chen et al., 2022; Gao et al., 2023; Zhao et al., 2025), and tool use (Schick et al., 2023; Hong et al., 2024; Wu et al., 2024). It is no exaggeration to assert that code corpus has fundamentally reshaped the landscape of modern LMs.

The advantages of integrating code into training corpora are manifold. First, models exposed to code exhibit superior performance in programming-related tasks (Ma et al., 2024; Aryabumi et al., 2025). Second, relative to heterogeneous Internet corpus, code is generally of higher quality and more structurally coherent, often leading to lower training loss at convergence (Figure 6). Third, the inherently structured nature of code enhances a model's ability to generate outputs that are logically organized and syntactically well-formed (OpenAI, 2024; Claude, 2024).

Beyond these evident benefits for programming tasks, an increasing body of research has begun to investigate whether code also confers advantages for non-coding abilities. For instance, Ma et al. (2024) examined the role of code corpus in both pretraining and supervised instruction tuning. Their

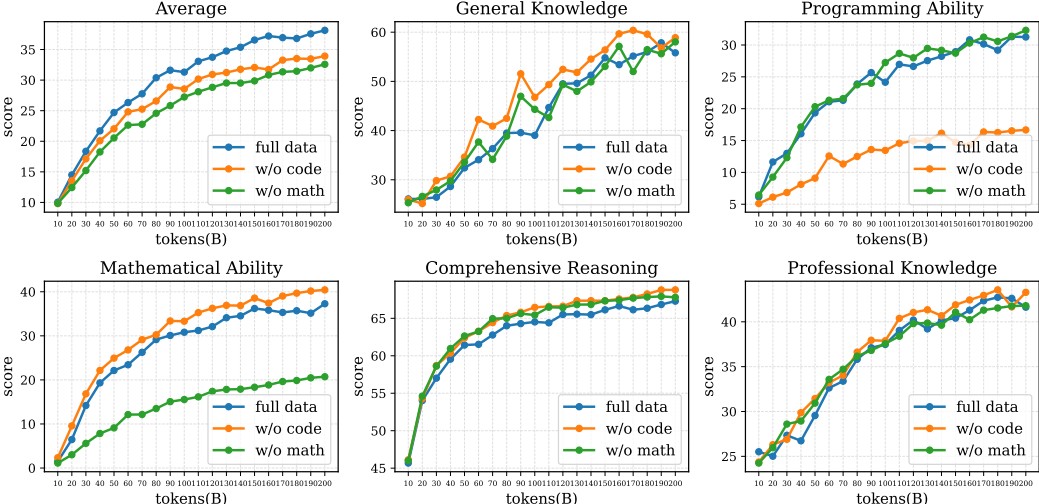

Figure 1: The impact of three distinct data compositions on model performance across multiple capability dimensions. Beginning with the 10T corpus, we conduct ablations by removing either code data (e.g., w/o code) or math data (e.g., w/o math), and subsequently evaluate the resulting models along five dimensions: general knowledge, coding ability, mathematical ability, comprehensive reasoning, and professional knowledge. Our findings reveal a clear competitive dynamic: code data competes with knowledge-intensive tasks, especially mathematical reasoning; whereas mathematical data competes with tasks requiring comprehensive, cross-domain reasoning.

findings indicate that mixed code–text pretraining substantially increases reasoning abilities, including logical, scientific, analogous, and legal domains, without inducing negative transfer. Aryabumi et al. (2025) further explored the influence of code quality and corpus composition on natural language reasoning, showing that allocating approximately 25% of the corpus to code strikes the optimal balance. They also reported that supplementing with high-quality synthetic code is more beneficial than relying solely on web-scraped sources. Despite methodological differences, both studies converge on a counterintuitive yet compelling conclusion: incorporating code data into the training corpus can enhance reasoning abilities in general-purpose LLMs.

In this paper, we revisit and critically reassess the counterintuitive claim that "code enhances reasoning ability." To this end, we design rigorous large-scale controlled experiments. We construct a 10T high-quality corpus and pretrain mixture of experts (MoE) models of varying sizes from scratch. On this foundation, we conduct two systematic ablation studies: (i) ablating the code corpus and (ii) ablating the math corpus. We find that the effects of different data corpora on model capabilities are not mutually reinforcing but rather exhibit a significant competitive relationship. Specifically, the code corpus primarily competes with knowledge-intensive tasks, whereas the math corpus competes mainly with comprehensive reasoning tasks. The inclusion of the code corpus leads to an overall performance degradation of 10.1% on downstream mathematical tasks, with a limited impact on foundational reasoning but a substantial negative effect on complex mathematical reasoning, where performance drops by as much as 71.53%. Conversely, incorporating the math corpus boosts performance on competitive programming tasks (by up to 37.11%) yet impairs the model's capabilities on code reasoning tasks (with a maximum decline of 17.30%).

We attribute the discrepancy between our findings and prior research to differing definitions of corpus types. Unlike previous approaches that treated code as a generalized category, our curated corpus adheres to stricter standards of quality and classification precision, explicitly distinguishing between 'pure code' and 'cross-domain' sources. In our framework, code is strictly defined as executable functions or program segments, excluding explanatory text or instructional comments. This delineation enables us to accurately distinguish between programming ability and cross-domain knowledge at the corpus level, thereby revealing their differential impacts on model capabilities with greater clarity. Consequently, our empirical study demonstrates that pure code data alone does not directly enhance reasoning abilities. Instead, the significant gains originate from mixed-reasoning data, which integrates code, natural language, and mathematical knowledge.

Building on this insight, we design a further experiment to validate the role of cognitive scaffolding. We curate a core set of structured reasoning data and deliberately increase its training proportion while keeping the overall mathematical training budget constant. The results revealed that performance on complex mathematical reasoning tasks significantly improved, while performance on code benchmarks remained almost unaffected (a negligible decrease of approximately 1%). This finding suggests that the structured reasoning signals function as a meta-reasoning scaffold within the model, playing a critical role in enhancing performance on highly difficult reasoning tasks.

Finally, we seek to explain the cohesive and adversarial relationships between domain data corpus by examining the model's internal mechanisms. Through an analysis of the expert routing distributions in the MoE model under various data configurations, and by leveraging hook mechanisms, we found that the cognitive scaffold significantly enhances complex reasoning capabilities while preserving the stability of the model's expert distribution. This result further corroborates its potential value as a cross-domain adhesive.

## 2 RELATED WORKS

### 2.1 DATA INFLUENCE ON LLM REASONING

Data constitute the fundamental source of model capabilities (Schaeffer et al., 2023; Razeghi et al., 2022). Understanding how data from different domains, particularly code and mathematics, interact, cooperate, or compete within LLMs has emerged as a central theme in recent research. Ma et al. (2024) investigated the impact of introducing code corpora at different training stages and concluded that pretraining on mixed code and text substantially enhances reasoning abilities (including logical, scientific, analogical and legal reasoning), with little evidence of negative transfer. Similarly, Aryabumi et al. (2025) demonstrated that incorporating high-quality synthetic code during both pretraining and annealing phases improves reasoning performance, while cautioning that an excessive proportion of code data (more than $3/4$) severely compromises performance on knowledge-intensive tasks. Collectively, these studies converge on the view that code plays an indispensable role in strengthening reasoning and generalization in LLMs.

In contrast, our work revisits this consensus from a critical perspective. By conducting experiments with a similar overarching design, but with deliberate modifications in key details, we arrive at a fundamentally different conclusion. Building on this divergence, we further identify the underlying factor that truly accounts for enhanced reasoning ability, which we term cognitive scaffolds.

### 2.2 DATA SELECTION AND DATA MIXING

Data selection and data mixing represent two complementary strategies for optimizing training corpora. These approaches operate at different levels of granularity: instance-level optimization and domain-level optimization, respectively. Data selection focuses on identifying the most valuable pretraining instances, thereby accelerating convergence and improving downstream performance (Albalak et al., 2024). Depending on when they are applied, selection methods can be categorized as offline or online. Offline methods filter corpora prior to training, using techniques such as data pruning (Marion et al., 2023; Tirumala et al., 2023) and data programming (Ratner et al., 2016; Zhou et al., 2024). Online methods, in contrast, dynamically adjust sampling strategies during training. For example, Gu et al. (2025) employed Pontryagin's Maximum Principle to prioritize high-quality data that provides the steepest descent in gradient-based optimization, while Jiang et al. (2025) estimated domain-level loss during training to preferentially sample more promising data, thereby improving learning efficiency.

Data mixing addresses the problem of determining optimal sampling ratios across domains to maximize overall model performance. DoReMi leverages group distributionally robust optimization to train a small proxy model without requiring prior knowledge of downstream tasks. The proxy model then generates domain weights that are subsequently used to resample large-scale corpora for LLMs training (Xie et al., 2023). Another line of work, such as REGMIX, trains meshed proxy models under varying data mixing strategies and employs regression to predict model performance across ratios, thereby inferring an optimal mixture strategy (Liu et al., 2025).

Building on these foundations, this paper integrates data selection and data mixing into a unified paradigm. On the one hand, adjusting mixture ratios enables a systematic exploration of cooperative and competitive dynamics across domains; on the other hand, the insights gained from this exploration feed back into the data selection stage, allowing us to identify and prioritize instances that exhibit synergistic effects across domains.

## 3  EXPERIMENT SETTINGS

### 3.1  MODEL ARCHITECTURE

For our experiments, we employ mixture-of-experts (MoE) models of varying scales. In this architecture, the standard feed-forward network (FFN) is replaced by a collection of $N$ experts(Fedus et al., 2022; Lepikhin et al., 2020; Jiang et al., 2024), each constituting a compact, modular FFN unit. This design enhances both the efficiency and the specialization of LLMs. The MoE model dynamically routes each token to a subset of experts via an individual router $R$, defined as follows:

$$
\begin{aligned}
g_t &= \mathrm{Softmax}(R(o_t)), \\
p_t &= \sum_i g_{t,i} E_i(o_t) \quad \mathrm{s.t.} \quad g_{t,i} \in \mathrm{TopK}(g_t),
\end{aligned}
\tag{1}
$$

where $o_t \in \mathbb{R}^d$ is the $d$-dimensional output of the multi-head attention, $E_i$ represents the $i$-th expert, $g \in \mathbb{R}^{\mathbb{N}}$ indicates the gating function, and $p_t$ denotes the output presentation of the $t$-th token.

To further improve training efficiency and scalability, we adopt a fine-grained expert strategy on top of the standard MoE (Dai et al., 2024; Liu et al., 2024). While maintaining a fixed total number of model parameters, we increase the number of experts proportionally while reducing the intermediate dimensionality of each expert. This configuration promotes greater specialization for each expert. To prevent any single expert from over-prioritizing general knowledge due to its reduced capacity, we introduce an additional shared expert $E_s$ trained on all tokens, thereby ensuring the presence of a universal expert. The entire procedure can be formally expressed as follows:

$$
p'_t = p_t + E_s(o_t).
\tag{2}
$$

An essential component of the MoE architecture is the routing module. In our design, we adopt a dropless routing strategy in conjunction with load-balancing loss and router z-loss to enhance training efficiency and prevent uneven token allocation across experts. To further mitigate instability during the early stages of pretraining, we introduce a novel mechanism termed stochastic routing warmup. This method injects controlled randomness into the routing process, thereby alleviating expert overloading and preventing expert collapse caused by severe routing imbalance in the initial training phase.

Formally, let $s_t \in \mathbb{R}^N$ denote the routing logits for an input token representation $h_t \in \mathbb{R}^{d'}$, computed by a linear projection layer. During the warm-up phase (i.e., when the current step $t_c$ less than warm-up step $t_w$), we interpolate between the learned logits and synthetic random logits. The final routing logits $s'_t$ are then given by:

$$
\begin{aligned}
s_t &= W^{\mathrm{T}} h_t + b, \\
s'_t &= \alpha s_t + (1 - \alpha)(\mu_s + \sigma_s \cdot \epsilon), \quad \epsilon \sim \mathcal{N}(0, 1),
\end{aligned}
\tag{3}
$$

where $W \in \mathbb{R}^{d' \times N}$ is a small projection layer, $\alpha = \min(\frac{t_c}{t_w}, 1.0)$ is warm-up gate, $\mu_s$ and $\sigma_s$ represent the running mean and standard deviation of $s_t$.

### 3.2  DATA PREPARATION AND DOMAIN DIVISION

We present a comprehensive and systematically constructed pretraining corpus of approximately 10T tokens, spanning six major domains: Web, Code, Math, Wikipedia, Books, and Multilingual. To ensure model quality, we have strict standardized processes for data acquisition, curation, and access. The design and construction of this pipeline are grounded in rigorous data engineering principles. We will introduce the methodological foundations that ensure the quality, balance, and utility of the dataset.

### 3.2.1 DATA ACQUISITION AND CURATION PIPELINE

The data pipeline comprises three stages: **collection**, **curation**, and **admission control**, designed to ensure high-quality, domain-balanced training inputs.

In the **collection phase**, raw data is sourced from diverse public repositories, including Common Crawl (Patel, 2020), GitHub (Hellendoorn & Sawant, 2021), arXiv (Clement et al., 2019), Project Gutenberg (Stroube, 2003), Wikipedia dumps (Foundation), and multilingual web archives (Braud et al., 2024). Code is collected from open-source repositories across major programming languages. Mathematical content draws from arXiv and educational platforms, augmented with internally generated synthetic datasets covering core concepts and proof patterns.

During **curation**, each data type undergoes targeted filtering to remove noise, structural errors, and low-signal content. Code files are validated for syntax, length, duplication, and functional density; math data is checked for LaTeX correctness, expression well-formedness, and explanatory coherence. All processing parameters and language-specific rules are detailed in Appendix A.2.

In the **admission control** stage, segments are scored using a fine-grained framework of over 300 metrics across 10 dimensions—including coherence and factual consistency. Domain-specific scoring criteria (e.g., algorithmic richness for code, derivation depth for math) yield tiered labels (high-/medium/low), which determine sampling weights in pretraining (Raffel et al., 2023). This enables prioritized learning from high-signal content while maintaining cross-domain balance.

### 3.2.2 DOMAIN CATEGORIZATION

The corpus is organized into six well-defined, semantically coherent domains, each serving distinct cognitive and linguistic functions in model learning. **Web**: General-purpose natural language from diverse online sources, providing breadth and real-world linguistic variation. **Code**: Programming scripts and software artifacts emphasizing syntactic precision and algorithmic logic. **Math**: Formal mathematical expressions, proofs, derivations, and problem-solution pairs that foster symbolic reasoning. **Wikipedia**: Structured encyclopedic knowledge with cross-referenced facts, supporting factual grounding and conceptual understanding. **Books**: Long-form narrative and expository texts that promote discourse coherence and deep semantic comprehension. **Multilingual**: High-quality parallel and monolingual texts across major world languages, enabling cross-lingual transfer.

This taxonomy avoids overlap and ambiguity by enforcing strict boundary definitions—particularly between hybrid sources (e.g., Jupyter notebooks) and pure-domain content. For instance, code embedded in explanatory prose (Code-NL) is separated from standalone code repositories, ensuring clean attribution and controlled mixing.

### 3.2.3 DATA ORGANIZATION AND MIXING STRATEGY

Data organization follows a two-phase strategy: *quality-tiered stratification* and *balanced mixture scheduling*. After cleaning and scoring, each domain is divided into high-, medium-, and low-quality strata. Only data above a defined threshold enters the final training mix.

During training, we employ a dynamic sampling policy that balances token-level contribution while prioritizing high-value domains such as Math and Code. Inspired by ablation studies on data contribution, our mixture ensures sufficient exposure to reasoning-intensive content without overwhelming general language distribution. The effectiveness of this strategy is validated through controlled experiments on small-scale models before large-scale deployment, ensuring stable convergence and optimal capability development.

## 4 EXPERIMENTAL OBSERVATIONS

We have constructed a 10T training corpus categorized into six domains: Web, Code, Math, Wikipedia, Books, and Multilingual. We begin by addressing a central question: *How do code and mathematics corpora influence overall performance?* To investigate this, we apply causal interventions by ablating specific data subsets from the full corpus to estimate their causal effect on downstream performance. Concretely, we counterfactually remove either Code or Math corpus and

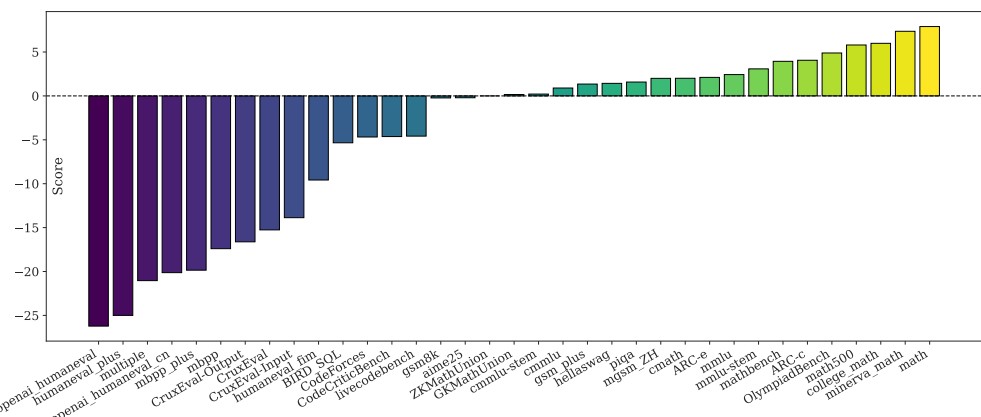

Figure 2: Code data exhibits a competitive relationship with knowledge-intensive tasks. When ablated from the full corpus, its absence leads to a pronounced decline in performance across all programming benchmarks, as expected. Beyond programming, code data also competes with comprehensive reasoning tasks such as PIQA and HellaSwag. For mathematical reasoning, however, the impact is more nuanced and task-dependent: code data significantly hinders performance on complex benchmarks (e.g., Math, OlympiadBench), while its effect on simpler problems (e.g., GSM8k) remains comparatively limited.

evaluate the resulting models across five capability dimensions: general knowledge, programming ability, mathematical ability, comprehensive reasoning, and professional knowledge.

Contrary to prior claims that "code data consistently enhances reasoning ability" (Ma et al., 2024; Aryabumi et al., 2025), our findings present a different picture. We observe that the interaction between domain-specific data and out-of-domain tasks can be characterized as a form of negative coupling. This phenomenon parallels interference in transfer learning, where optimization signals from one domain inadvertently impede knowledge acquisition in another. From a physical standpoint, it resembles a counteracting force: just as an action in one direction produces an opposing reaction that resists motion elsewhere, the inclusion of code data suppresses progress on knowledge-intensive tasks, while the presence of mathematical data counteracts improvements in comprehensive reasoning. Such negative coupling underscores a fundamental trade-off in multi-domain training: advances in one dimension are frequently attained at the expense of another, reflecting the delicate balance inherent in allocating representational capacity and optimization budget.

## 4.1 CODE DATA COMPETES WITH KNOWLEDGE-INTENSIVE TASKS

We evaluate performance differences between MoE models trained on the full corpus and those trained with code data ablated, with results presented in Figure 2. The experiments reveal a clear intra-domain coupling effect within code data: ablating code markedly diminishes the programming ability of LLMs, as expected. Beyond programming, code also introduces a resource competition effect for knowledge-intensive tasks. For instance, on comprehensive reasoning benchmarks such as PIQA and HellaSwag, including code reduces performance by 2.11% and 2.39%, respectively.

For mathematical reasoning, the competitive effect of code is more pronounced and strongly dependent on task difficulty. On average, code data reduces mathematical ability by 14.38%, although the impact varies across individual benchmarks. For relatively simple reasoning tasks, such as GSM8k (+0.4%) and CMATH (-3%), the negative effect is minor. For more complex benchmarks, the detrimental impact is substantial: Minerva-Math (-71.53%), OlympiadBench (-47.16%), Math (-22.64%), CollegeMath (-12.48%) and MathBench (-10.23%) all exhibit performance degradation.

## 4.2 MATH DATA COUNTERACTS WITH COMPREHENSIVE REASONING

Similar to code corpus, ablating math corpus weakens model performance within its own domain. However, in contrast to the strong competitive effect that code exerts on mathematics, the influence of mathematics on programming is less pronounced and instead reveals divergent trends across programming tasks. On programming competition benchmarks, incorporating math corpus

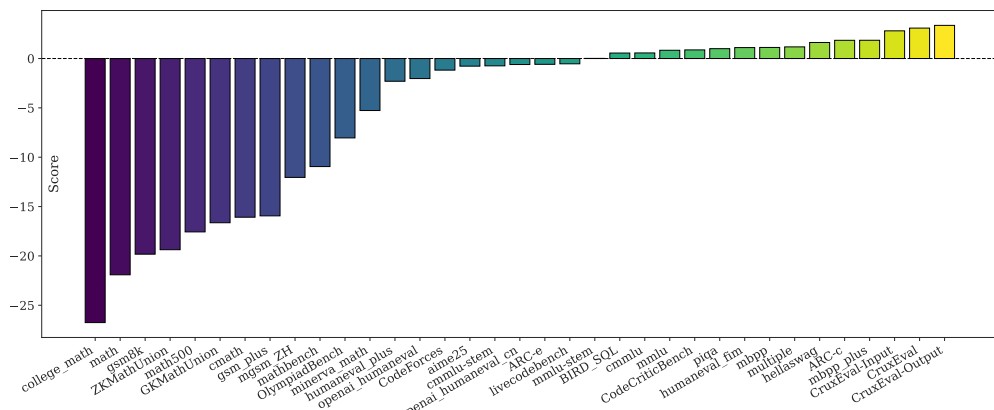

Figure 3: Math data exhibits a competitive effect on comprehensive reasoning tasks. As with code, ablating math data results in a pronounced decline on mathematical benchmarks. However, unlike code, mathematics exerts little competitive influence on programming ability. Instead, its competition manifests more prominently in comprehensive reasoning tasks, encompassing both code reasoning (e.g., CruxEval, MBPP) and commonsense reasoning (e.g., HellaSwag).

markedly improves performance (e.g., CodeForces +37.11%, LiveCodeBench +11.26%), suggesting that mathematical knowledge serves as a form of cognitive scaffolding, enabling models to better comprehend and solve programming problems with complex logical structures and algorithmic characteristics. By contrast, on code reasoning tasks that demand hybrid reasoning schema, mathematical data exhibits a clear antagonistic effect (e.g., CruxEval -17.30%, MBPP -6.12%). A similar negative trend is also observed on commonsense reasoning benchmarks (e.g., HellaSwag -2.94%), where the inclusion of math data results in measurable degradation.

**Why does this apparent inconsistency arise?** At first glance, our conclusions may seem to contradict prior studies, but in fact, the two are not necessarily in conflict. During our data preparation, we explicitly distinguish pure code data from Code–NL data, retaining the latter in our code ablation experiments. In other words, through careful control of variables, we effectively decouple the sources of model capabilities into "pure code" versus "cross-domain" data.

It is important to note that, although this cross-domain data is collected from programming-related websites, it is not inherently code-centric. For instance, Aryabumi et al. (2025) markup languages such as Markdown, CSS, and HTML were categorized as code for training purposes. However, these data primarily originate from web text and are only presented in a structured format. As a result, they contain substantial non-code knowledge, which explains why they reported improvements across multiple other domains attributed to code data.

Based on this analysis, we conclude that, while data from different domains generally exhibits negative coupling, cross-domain intersections exist. Properly leveraging these intersections can partially mitigate the competitive effects between domains, offering opportunities to enhance multi-domain model performance.

## 5   COGNITIVE SCAFFOLDING FOR COMPLEX MATHEMATICAL REASONING

Building on the above observations and conclusions, we seek to identify cognitive scaffolds that can substantially enhance complex mathematical reasoning. Such scaffolds can be regarded as a core subset of mathematical corpora which, without competing against data from other domains, effectively improve models' reasoning performance on demanding tasks. Cognitive scaffolds serve as the abstract reasoning supports upon which models rely when tackling cross-domain problems. They not only foster generalization over symbolic and logical structures but also establish transferable frameworks for multi-step reasoning, causal analysis, and cross-domain comprehension. In other words, cognitive scaffolds do not merely boost memory or pattern matching for a specific task; rather, by laying down a general foundation for reasoning, they enable models to perform logical inference and knowledge integration more effectively in complex contexts.

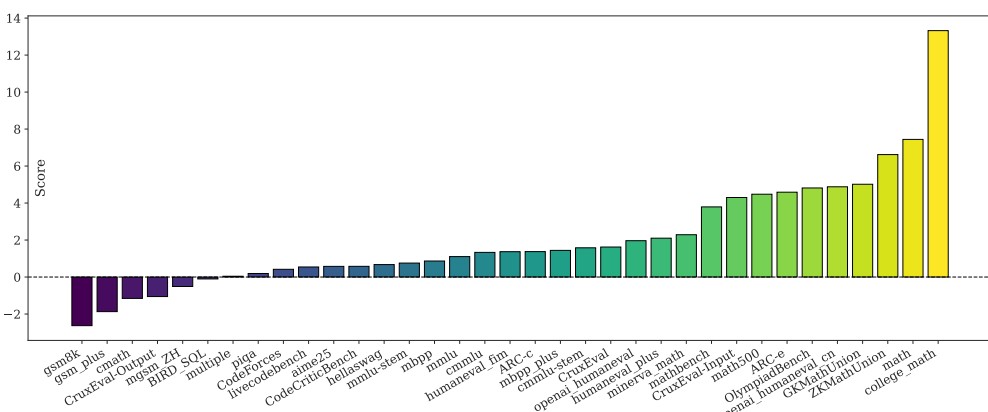

Figure 4: Incorporating structured reasoning data enhances model performance on different tasks. For challenging datasets such as College Math and MATH, the introduction of cognitive scaffolds promotes generalization and thus improves model effectiveness. By contrast, for tasks that can be solved without structured reasoning, such as GSM8k and CMath, the addition of such data introduces competition and may even hinder performance.

We observe that cognitive scaffolds exhibit the hallmarks of "structured reasoning", Formally, such data are highly structured, often characterized by explicit symbolic systems, hierarchical derivation procedures, and rigorous logical chains, thereby providing models with transparent reasoning trajectories and traceable intermediate steps. In terms of content, these data are closely aligned with mathematical reasoning, encompassing paradigms such as formula derivation, proof construction, equation solving, and function modeling. This demands not only precise symbolic manipulation at the local level but also logical coherence and correct cross-step dependencies at the global level.

To retrieve scaffold-like data, we trained a lightweight FastText model (Bojanowski et al., 2017) on roughly 400,000 samples, identifying around 200,000 structured positives (mainly from code-related corpora) and collecting negatives from web corpora filtered via regex and symbol-density statistics. Training on a balanced mix of positive and negative samples enhanced generalization in detecting structured reasoning data. For further details regarding the selection criteria of additional cognitive scaffolds, please refer to Appendix A.6.

We then incorporated these structured reasoning data into training while keeping the overall proportion of mathematical corpora unchanged (i.e., under the same budget for math-domain data). The experimental results are presented in Figure 4. They show that cognitive scaffolds yield particularly notable improvements on complex reasoning tasks: on average, overall mathematical reasoning performance increased by 17.56%. However, this benefit came with a trade-off: on relatively simple tasks (e.g., GSM8k and CMath), structured reasoning disrupted cases that could otherwise be solved directly via natural language, leading to performance drops of 6.29% and 2.00%, respectively. Nevertheless, these losses are acceptable when weighed against the substantial gains on complex tasks. For instance, significant improvements were observed on College Math (+30.05%), MATH (+23.17%), OlympiadBench (+47.78%), and MathBench (+14.51%). These results underscore the pivotal role of structured reasoning in complex mathematical problem-solving, while also highlighting its competitive interaction with unstructured (natural language) reasoning.

Beyond mathematics, scaffold-like data possess an inherent potential for cross-domain transfer. Their embedded frameworks of logical deduction and abstract pattern recognition can equally benefit non-mathematical scenarios such as program verification, experimental design, and causal modeling. Thus, structured reasoning data are not only key enablers of mathematical reasoning but also provide a broadly applicable paradigm for addressing complex tasks across diverse domains.

## 6  INSIGHTS FROM MOE ROUTING DISTRIBUTION

Finally, we seek to interpret the cohesion and antagonism among different data sources from the perspective of the internal mechanisms of the model. To this end, we analyze the expert routing distributions of a MoE model under varying data configurations during auto-regressive generation across domains. Specifically, we examine the distribution of activated experts in Math (GSM8k,

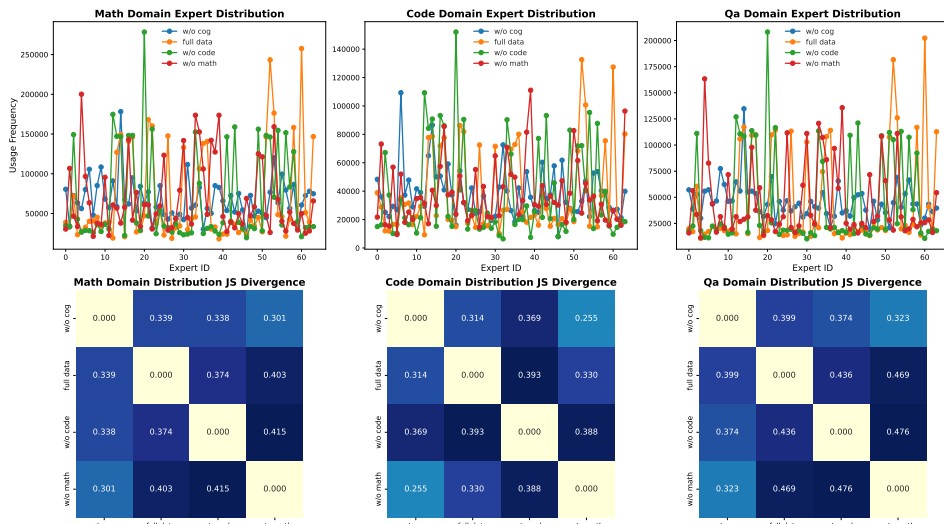

Figure 5: MoE Expert routing distribution and JS divergence in Math, Code, and QA domains under different data configurations.

MATH500), Code (HumanEval, LiveCodeBench), and Question Answering (MMLU), and quantify distributional discrepancies using Jensen–Shannon (JS) divergence (Menéndez et al., 1997). The results are presented in Figure 5.

Our analysis reveals that, across all domains, ablating the corresponding domain-specific data results in the most pronounced distributional shifts. For instance, in the Code domain, the divergence of the 'w/o code' configuration is substantially larger than that of other settings; a similar pattern is observed in the Math domain with the 'w/o math' configuration. This demonstrates that domain-specific data exert a highly specialized influence on internal representations and expert activation, and their removal disrupts the original distributional balance.

In contrast, the inclusion of cognitive scaffolds (e.g., w/o cog) induces the smallest distributional drift across all domains. This suggests that their primary contribution lies not in reshaping domain-specific representations, but in supporting cross-domain reasoning. In other words, cognitive scaffolds enhance complex reasoning while preserving stability in expert activation distributions, highlighting their potential role as a cross-domain adhesive within LLMs.

# 7 CONCLUSION AND FUTURE WORK

In this work, we rethought the counterintuitive claim that "code corpus enhances reasoning ability" through controlled experiments and fine-grained domain division. We established a standardized pipeline for data collection and cleaning, yielding a 10T-token corpus constructed under strict data admission policies. Focusing on two key domains: Code and Math, we design corpus configurations with high intra-domain cohesion and low inter-domain coupling, and systematically evaluate their effects through ablation studies. Our results show that code corpora exert strong competitive effects on knowledge-intensive tasks, particularly complex mathematical reasoning, while mathematical corpora interfere more with comprehensive reasoning. Based on these observations, we identify cognitive scaffolds that play a pivotal role in supporting complex reasoning. Incorporating such data yields substantial improvements in complex mathematical reasoning under conditions of low cross-domain competition.

Finally, by analyzing shifts in expert activation distributions across different corpus configurations, we provide mechanistic insights into how data composition shapes the internal dynamics of LLMs, giving rise to either competitive or synergistic effects across domains. Taken together, our findings highlight a promising path for data-centric model optimization: Under fixed budgets, appropriately including cross-domain synthetic data such as structured reasoning corpora can mitigate the "seesaw effect", preserving performance in one domain while enhancing it in another.

## 8 ETHICS STATEMENT

This work adheres to the ICLR Code of Ethics. In this study, no human subjects or animal experimentation was involved. All datasets used were sourced in compliance with relevant usage guidelines, ensuring no violation of privacy. We have taken care to avoid any biases or discriminatory outcomes in our research process. No personally identifiable information was used, and no experiments were conducted that could raise privacy or security concerns. We are committed to maintaining transparency and integrity throughout the research process.

## 9 REPRODUCIBILITY STATEMENT

We have made every effort to ensure that the results presented in this paper are reproducible. All code and datasets have been made publicly available in an anonymous repository to facilitate replication and verification. The experimental setup, including training steps, model configurations, and hardware details, is described in detail in the paper. We also provided the complete training process in Appendix A.3 to ensure the authenticity of our training.

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

# A APPENDIX

## A.1 USE OF LLMs

Large Language Models (LLMs) were used to aid in the writing and polishing of the manuscript. Specifically, we used an LLM to assist in refining the language, improving readability, and ensuring clarity in various sections of the paper. The model helped with tasks such as sentence rephrasing, grammar checking, and enhancing the overall flow of the text.

It is important to note that the LLM was not involved in the ideation, research methodology, or experimental design. All research concepts, ideas, and analyses were developed and conducted by the authors. The contributions of the LLM were solely focused on improving the linguistic quality of the paper, with no involvement in the scientific content or data analysis.

The authors take full responsibility for the content of the manuscript, including any text generated or polished by the LLM. We have ensured that the LLM-generated text adheres to ethical guidelines and does not contribute to plagiarism or scientific misconduct.

## A.2 DATA PROCESSING DETAILS

This appendix provides comprehensive details on the data curation pipeline described in Section 3.2.1, including code filtering rules, mathematical content validation procedures, and synthetic data generation protocols. These components are designed to ensure high signal-to-noise ratio and structural integrity across domains.

### A.2.1 CODE DATA CLEANING RULES

To maintain high-quality code inputs, we apply a multi-stage, language-specific filtering framework. The following table summarizes the core cleaning rules applied to source files collected from GitHub and other public repositories.

Table 1: Code cleaning rules by programming language.

| Filter Type | Python | JavaScript | Java | C++ | General |
|---|---|---|---|---|---|
| Long line filter | > 200 chars per line | > 150 chars per line | > 150 chars per line | > 150 chars per line | Apply soft wrap detection |
| File length filter | < 10 or > 10k tokens | < 10 or > 8k tokens | < 15 or > 9k tokens | < 15 or > 9k tokens | Exclude stubs and artifacts |
| Syntax validation | AST parsing via `libcst` | ESTree AST (via `espree`) | Javalang parser | Tree-sitter | Reject invalid syntax |
| Format check | PEP8 compliance check | ESLint basic rules | Check braces/indentation | GCC pre-parse | Remove malformed layout |
| Low-quality language filter | Filter docstrings-only | Filter HTML-in-JS (e.g., JSX only) | Filter interface-only files | Filter header-only if trivial | Remove boilerplate-heavy files |
| Language-specific filters | No `.pyc`, `__pycache__` | No minified (`.min.js`) | No auto-generated (Lombok, etc.) | No generated bindings | Block known bad patterns |
| Signal density threshold | < 30% executable code (after comment removal) → discard | Same rule applied across all languages | | | |
| Deduplication level | Function-level (via normalized AST) and repo-level (SimHash) | Retain canonical implementation | | | |

Key definitions:

- **Executable code**: lines containing function bodies, control flow statements, or expressions; excludes imports, comments, type annotations, and empty lines.

- **Normalized AST**: abstract syntax trees with identifiers anonymized and constants replaced to detect semantic duplicates.

- **Signal density**: ratio of executable code lines to total file size after preprocessing.

All filtering steps are executed sequentially. Files failing any critical rule (syntax, format, signal density) are discarded. Surviving files are further deduplicated before inclusion in the training corpus.

### A.2.2 MATHEMATICAL CONTENT VALIDATION

For mathematical data sourced from arXiv and internal generators, we enforce strict structural and semantic checks to preserve correctness and pedagogical value.

**Validation Pipeline**

The processing pipeline for math content follows these stages:

1. **Source normalization**: Convert PDFs and HTML to structured text using `GROBID` and custom LaTeX extractors.

2. **LaTeX syntax check**: Validate all math environments ($, $$, `equation`, `align`) using `LaTeXML` and regex-based grammar rules.

3. **Expression well-formedness**: Detect unbalanced parentheses, mismatched delimiters, undefined commands, and incomplete integrals/sums.

4. **Consistency check**: Ensure equation labels are referenced, theorem-environment nesting is correct, and variables are consistently used.

5. **Noise detection**: Flag segments with excessive placeholders, repeated derivations, or formula-to-text ratio exceeding 80%.

6. **Final filtering**: Remove entries failing two or more checks above.

### A.3 TRAINING DETAILS

We pretrain a 20-layer autoregressive MoE language model with a hidden size of 2048 and 16 attention heads. Each MoE layer contains 16 experts with an intermediate size of 2048, and the feed-forward network (FFN) hidden size in dense layers is set to 5120. The MoE router selects the top-2 experts per token and employs a sigmoid score function with a router bias update rate of 0.001.

Training is conducted using a pipeline-parallel and tensor-parallel setup with global batch size of 2048 and micro-batch size of 4. Sequence parallelism and gradient accumulation fusion are enabled to optimize memory usage. The model is trained in bfloat16 and FP8 mixed precision.

Optimization is performed with AdamW ($\beta_1 = 0.9, \beta_2 = 0.95$, weight decay=0.1) and a constant learning rate of $5 \times 10$, with 2,000 warmup iterations. Training uses a block LM objective with masked softmax fusion, Swiglu activation, and unidirectional attention. The embedding weights and output projection are untied, and a norm head is not used. Training runs for 24,000 iterations. Model checkpoints are saved every 1,200 iterations.

### A.4 DETAILED EXPERIMENTAL RESULTS

Detailed experimental data are provided in Table 2. The results indicate a significant competitive relationship between code corpus and mathematical corpus. In contrast, while competitive relationships exist in general knowledge, comprehensive reasoning, and professional knowledge, they are not statistically significant.

### A.5 THE SAME CONCLUSION WAS MAINTAINED ACROSS DIFFERENT ARCHITECTURES AND SIZES.

To validate the robustness and consistency of our conclusions, we trained two additional model variants using identical configurations to our original setup: (1) We replaced the MoE architecture with dense models at 1B and 5B scales, maintaining consistent training data and computational budget. And (2) we adjusted the total parameter count in the MoE architecture by varying the number of experts, by reducing from 64 to 32 (32e) or 16 experts (16e).

As summarized in Table 2, the results consistently align with those reported in the main text, indicating that architectural choices have minimal impact on model capabilities and primarily affect training and inference efficiency. Instead, model performance is predominantly determined by three key factors: parameter count, data volume, and computational budget.

At a finer granularity, we also observe a cross-domain scaling law: under a fixed total budget, allocating more resources to the code domain correspondingly reduces the budget for mathematical domains, leading to performance degradation in the latter.

### A.6 DETAILED SELECTION CRITERIA OF COGNITIVE SCAFFOLDS

The objective of cognitive scaffolding is to identify segments exhibiting "structured reasoning patterns" from large-scale math corpora. To this end, we developed a foundational binary classifier capable of distinguishing between "code-structured structures" and "unstructured text." A meticulously defined training set was constructed as follows:

Table 2: Detailed results under different architectures and model parameter settings.

### MATHEMATICAL ABILITY

| Model | gsm8k | cmath | minerva_math | aime25 | math500 | OlympiadBench | GKMathUnion | ZKMathUnion | Overall |
|---|---|---|---|---|---|---|---|---|---|
| full data | 58.00↑ | 67.94 | 10.29 | 2.92↑ | 31.00 | 10.37 | 39.05 | 62.02↑ | 37.26 |
| w/o code | 57.77 | 69.95↑ | 17.65↑ | 2.71 | 36.80↑ | 15.26↑ | 39.21↑ | 62.02↑ | 40.43↑ |
| w/o math | 33.51 | 50.46 | 7.35 | 0.00 | 10.60 | 5.88 | 18.41 | 42.28 | 20.71 |
| full data (32e) | 54.59 | 65.39 | 9.93 | 2.92↑ | 26.40 | 10.22↑ | 41.59↑ | 59.35 | 36.20 |
| w/o code (32e) | 56.48↑ | 70.13↑ | 13.60↑ | 1.67 | 33.60↑ | 9.33 | 36.51 | 65.73↑ | 38.52↑ |
| w/o math (32e) | 25.25 | 47.81 | 4.04 | 0.00 | 7.00 | 1.19 | 19.37 | 38.87 | 17.71 |
| full data (16e) | 46.40 | 59.93↑ | 9.93 | 0.42 | 24.00 | 14.22↑ | 31.59 | 56.53 | 31.60 |
| w/o code (16e) | 49.73↑ | 59.29 | 13.24↑ | 1.04↑ | 27.00↑ | 10.81 | 31.90↑ | 57.27↑ | 32.91↑ |
| w/o math (16e) | 22.29 | 42.90 | 5.15 | 0.42 | 8.80 | 2.37 | 14.29 | 36.20 | 15.99 |
| full data (5B) | 64.59↑ | 73.04 | 18.01↑ | - | 38.00 | 15.56↑ | 23.65 | 64.54↑ | 45.13 |
| w/o code (5B) | 63.08 | 73.68↑ | 14.34 | - | 45.40↑ | 12.91 | 33.97↑ | 63.95 | 46.56↑ |
| w/o math (5B) | 44.28 | 57.19 | 6.25 | - | 11.60 | 2.67 | 20.48 | 45.99 | 26.71 |
| full data (1B) | 26.99 | 45.17 | 7.35↑ | - | 22.80 | 4.15↑ | 20.00 | 42.58 | 20.08 |
| w/o code (1B) | 29.57↑ | 48.27↑ | 6.99 | - | 25.60↑ | 0.59 | 21.60↑ | 44.96↑ | 25.11↑ |
| w/o math (1B) | 10.99 | 30.42 | 2.57 | - | 3.00 | 2.07 | 6.40 | 23.44 | 9.66 |

### PROGRAMMING ABILITY

| Model | humaneval | humaneval_fim | mbpp | mbpp_plus | CruxEval | CodeCriticBench | CodeForces | BIRD_SQL | Multipl-E | LiveCodeBench | Overall |
|---|---|---|---|---|---|---|---|---|---|---|---|
| full data | 46.34↑ | 59.63 | 31.60 | 40.48 | 25.38 | 56.23↑ | 7.54↑ | 5.80↑ | 31.05 | 6.54↑ | 31.25 |
| w/o code | 20.12 | 50.05 | 14.20 | 20.63 | 10.12 | 51.60 | 2.86 | 0.46 | 10.01 | 1.96 | 16.67 |
| w/o math | 45.12 | 61.57↑ | 35.80↑ | 42.86↑ | 31.00↑ | 55.00 | 1.17 | 5.80↑ | 32.87↑ | 5.88 | 32.30↑ |
| full data (32e) | 46.95↑ | 59.05 | 33.00 | 36.77 | 24.81↑ | 50.72 | 5.62↑ | 4.01 | 28.12 | 4.90 | 26.94↑ |
| w/o code (32e) | 17.68 | 46.47 | 14.20 | 17.46 | 10.56 | 53.49 | 1.17 | 0.49 | 7.20 | 0.65 | 14.25 |
| w/o math (32e) | 38.41 | 60.31↑ | 34.80↑ | 39.68↑ | 21.12 | 54.33↑ | 3.93 | 5.18↑ | 29.00↑ | 5.56↑ | 24.25 |
| full data (16e) | 57.89 | 37.80↑ | 39.15↑ | 34.15↑ | 19.06↑ | 55.35↑ | 1.17↑ | 3.29 | 25.62↑ | 6.05↑ | 26.94↑ |
| w/o code (16e) | 46.76 | 13.41 | 14.29 | 12.80 | 9.06 | 52.09 | 1.17↑ | 0.29 | 5.66 | 0.49 | 14.25 |
| w/o math (16e) | 58.86↑ | 37.20 | 34.13 | 31.10 | 14.25 | 52.70 | 1.17↑ | 4.73↑ | 25.04 | 5.39 | 24.25 |
| full data (5B) | 48.78 | 63.02↑ | 42.68 | 45.50 | 35.94↑ | 56.53↑ | 1.17 | 9.81↑ | 28.78↑ | 8.50 | 35.38↑ |
| w/o code (5B) | 25.00 | 52.76 | 20.12 | 22.75 | 28.06 | 53.19 | 3.93 | 2.71 | 9.08 | 4.58 | 23.14 |
| w/o math (5B) | 49.39↑ | 62.15 | 45.12↑ | 45.77↑ | 34.94 | 52.47 | 5.85↑ | 6.94 | 17.97 | 9.31↑ | 34.10 |
| full data (1B) | 24.39↑ | 51.11 | 19.20↑ | 21.34↑ | 13.38↑ | 52.81 | 1.17↑ | 1.79↑ | 14.92 | 0.65 | 18.71↑ |
| w/o code (1B) | 2.44 | 39.79 | 4.00 | 1.83 | 5.44 | 53.26 | 1.17↑ | 0.13 | 2.39 | 0.00 | 9.09 |
| w/o math (1B) | 9.15 | 51.89↑ | 18.00 | 7.93 | 11.94 | 53.77↑ | 1.17↑ | 1.56 | 15.78↑ | 1.31↑ | 15.99 |

### GENERAL KNOWLEDGE

| Model | ARC-c | ARC-e | Overall |
|---|---|---|---|
| full data | 44.41 | 67.20 | 55.81 |
| w/o code | 48.47↑ | 69.31↑ | 58.89↑ |
| w/o math | 48.14 | 67.90 | 58.02 |
| full data (32e) | 41.02 | 64.37 | 52.29 |
| w/o code (32e) | 48.81↑ | 66.49↑ | 53.48 |
| w/o math (32e) | 43.73 | 62.79 | 53.89↑ |
| full data (16e) | 43.73 | 60.85↑ | 52.29 |
| w/o code (16e) | 46.10 | 60.85↑ | 53.48 |
| w/o math (16e) | 47.46↑ | 60.32 | 53.89↑ |
| full data (5B) | 58.31 | 74.96 | 66.64 |
| w/o code (5B) | 61.02↑ | 78.84↑ | 69.93↑ |
| w/o math (5B) | 60.00 | 77.78 | 68.89 |
| full data (1B) | 28.47 | 34.04 | 31.26 |
| w/o code (1B) | 36.27↑ | 47.44↑ | 41.86↑ |
| w/o math (1B) | 34.24 | 43.74 | 38.99 |

### COMPREHENSIVE REASONING

| Model | piqa | hellaswag | Overall |
|---|---|---|---|
| full data | 47.98 | 74.86 | 67.29 |
| w/o code | 53.97↑ | 76.44↑ | 68.80↑ |
| w/o math | 24.24 | 74.54 | 67.81 |
| full data (32e) | 50.21 | 73.34 | 64.83 |
| w/o code (32e) | 54.33↑ | 74.97↑ | 66.34↑ |
| w/o math (32e) | 20.23 | 74.05 | 65.55 |
| full data (16e) | 73.45 | 56.20 | 64.83 |
| w/o code (16e) | 74.81↑ | 57.86↑ | 66.34↑ |
| w/o math (16e) | 73.50 | 57.60 | 65.55 |
| full data (5B) | 76.12↑ | 61.60 | 68.86 |
| w/o code (5B) | 76.12↑ | 62.82 | 69.47 |
| w/o math (5B) | 75.95 | 63.04↑ | 69.50↑ |
| full data (1B) | 70.51 | 47.41 | 58.96 |
| w/o code (1B) | 71.60 | 48.92 | 60.26 |
| w/o math (1B) | 72.25↑ | 48.96↑ | 60.61↑ |

### PROFESSIONAL KNOWLEDGE

| Model | cmmlu | cmmlu-stem | mmlu | mmlu-stem | Overall |
|---|---|---|---|---|---|
| full data | 42.39 | 37.27 | 46.05 | 40.80 | 41.63 |
| w/o code | 43.29 | 37.48↑ | 48.48↑ | 43.88↑ | 43.28↑ |
| w/o math | 43.55↑ | 36.85 | 47.04 | 39.74 | 41.80 |
| full data (32e) | 35.90 | 39.18 | 43.22 | 44.93 | 39.06 |
| w/o code (32e) | 37.36↑ | 40.42↑ | 42.72 | 45.43↑ | 39.16 |
| w/o math (32e) | 36.54 | 36.44 | 43.48↑ | 43.48 | 39.80↑ |
| full data (16e) | 38.77 | 35.01 | 43.22 | 39.23 | 39.06 |
| w/o code (16e) | 40.57 | 36.10↑ | 42.72 | 37.26 | 39.16 |
| w/o math (16e) | 41.18↑ | 35.09 | 44.38↑ | 38.54 | 39.80↑ |
| full data (5B) | 47.51 | 41.16 | 52.03 | 45.86 | 46.64 |
| w/o code (5B) | 48.98↑ | 42.64↑ | 53.84↑ | 48.59↑ | 48.51↑ |
| w/o math (5B) | 46.71 | 38.69 | 52.31 | 45.31 | 45.76 |
| full data (1B) | 32.13 | 28.82 | 33.85 | 29.61 | 31.10 |
| w/o code (1B) | 35.97↑ | 32.09↑ | 39.53↑ | 35.32↑ | 35.73↑ |
| w/o math (1B) | 31.13 | 27.95 | 33.92 | 31.06 | 31.02 |

### A.6.1 POSITIVE SAMPLE ANNOTATION PROTOCOL

Positive samples were sourced from our rigorously curated high-quality code dataset (approximately 200k instances). These samples, entirely devoid of mathematical content, demonstrate explicit structured logic and formal expressions (e.g., indentation rules, identifier patterns, operator sequences, function structures). Tree-sitter was employed to ensure all code samples are compilable.

All positive samples originated from publicly available, reproducible code sources (GitHub repositories, online judge datasets, and internally compliant open-source mirrors), with deduplication and near-duplicate removal applied.

### A.6.2 NEGATIVE SAMPLE CONSTRUCTION

Negative samples were drawn from extensively covered web text data. To ensure the absence of code-like structures, we applied lenient yet effective heuristics via regex filtering to exclude "code-structured" elements, including:

- Removing texts containing high-frequency code symbol patterns (e.g., =, ::, int);
- Filtering out segments with multi-line indentation, consecutive , or grouped operators;
- Excluding texts containing programming language keywords, code file extensions, or Markdown formatting.

Only natural language-dominated, syntactically loose texts were retained, ensuring broad coverage of negative samples without introducing misclassified "weakly-structured texts."

All data must be cleaned according to Appendix A.2 before being allowed access.

### A.6.3 TRAINING AND THRESHOLD SETTING PRINCIPLES

We collected around 400k training samples, and 188678 validation samples. A lightweight structure identifier was trained using FastText. The classification threshold was determined by: (1) Maximizing the F1-score on the validation set; (2) Prioritizing precision in identifying positive (structured) classes to avoid misclassifying ordinary mathematical text as "structured reasoning";

### A.6.4 CONTAMINATION AUDIT

All positive samples were code-based, with no overlap to mathematical data; Negative samples were sourced from web-based natural language, distinct from mathematical training corpora and code-related data;

The classifier was solely used to detect "structured reasoning patterns", without learning mathematical content itself; A sampled audit of the filtered cognitive scaffolding data confirmed the absence of residual code segments or unintended formatting contamination. Thus, the filter introduces no information leakage regarding mathematical content and ensures fairness in subsequent mathematical evaluations.

### A.6.5 CLASSIFIER PERFORMANCE AND CALIBRATION

The FastText classifier was comprehensively evaluated on the validation set, yielding: Accuracy: 0.9696, Positive Class Precision: 0.9998 and Positive Class Recall: 0.9665. In constructing cognitive scaffolds, we prioritized the optimization of precision over recall, as false positives would significantly contaminate the structured reasoning data and undermine their efficacy in enhancing mathematical reasoning. To maximize structural consistency and interpretability of the data, we adopted a conservative classification threshold, enabling the classifier to achieve high precision on the validation set while maintaining sufficient recall.

### A.6.6 MISCLASSIFICATION ANALYSIS

**False Positives.** Primarily originate from web texts with "pseudo-structured" formats, such as heavily bulleted lists, pseudo-code layouts, or Markdown tables. Although minimal, these reflect the model's sensitivity to structural cues.

**False Negatives.** Mostly involve weakly-structured code snippets embedded in long texts (e.g., single-line function prototypes or unindented expressions). These have negligible impact as our scaffolding construction emphasizes high precision — False Negatives cases merely reduce candidate scaffolds without introducing erroneous structures.

### A.7 TRAINING PROCESS RECORDS

We provide detailed training processes in various fields in the Figure below.

### A.8 DETAILED DOWNSTREAM PROFORMANCE

We show the evaluation results of the entire list in Figure 16.

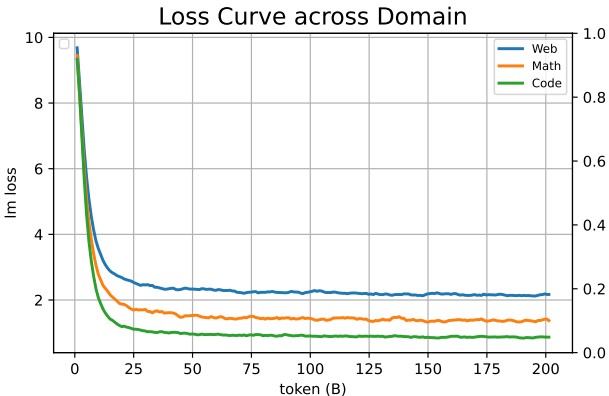

Figure 6: The loss in each domain decreases during training. It can be seen that the loss in the code domain decreases the fastest, and the convergence value is also lower.

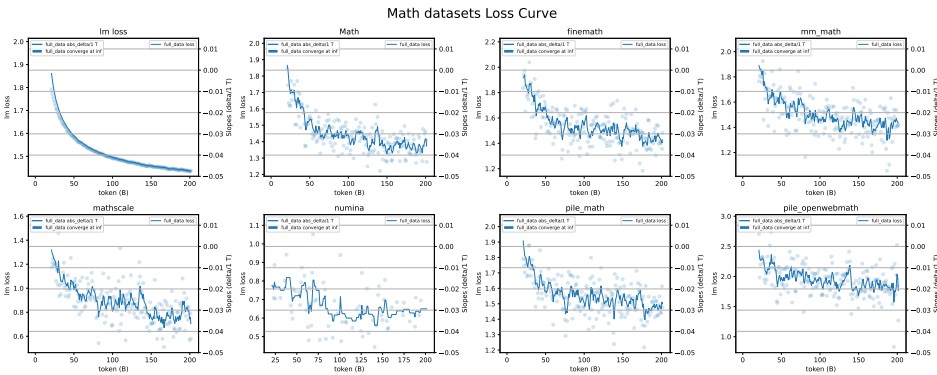

Figure 7: The loss in math decreases during training.

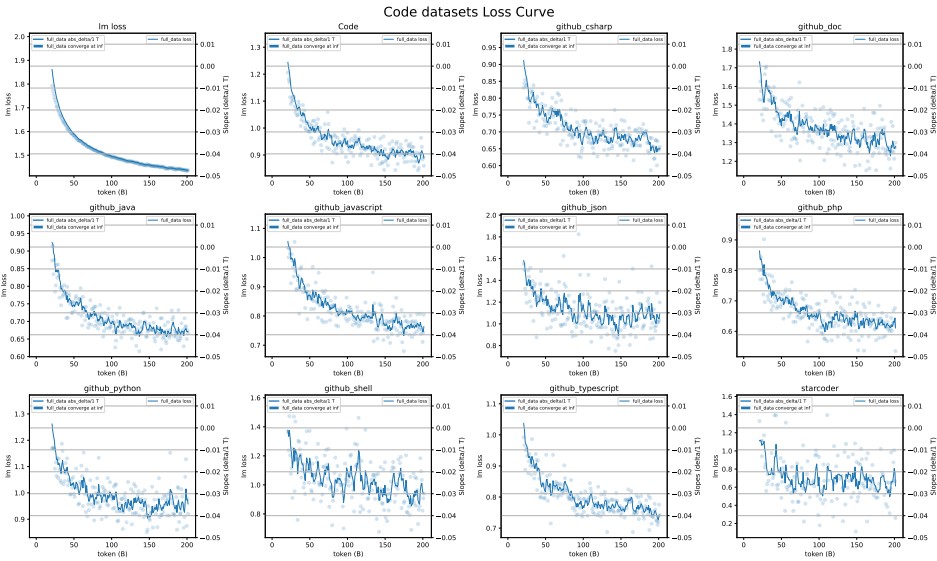

Figure 8: The loss in math decreases during training.

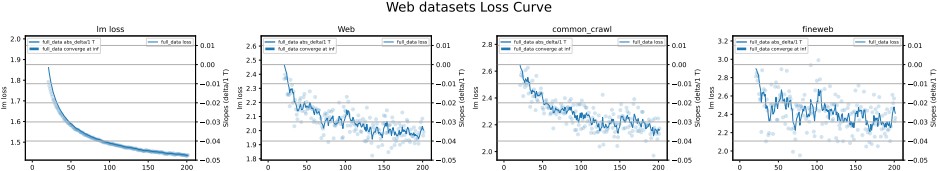

Figure 9: The loss in web decreases during training.

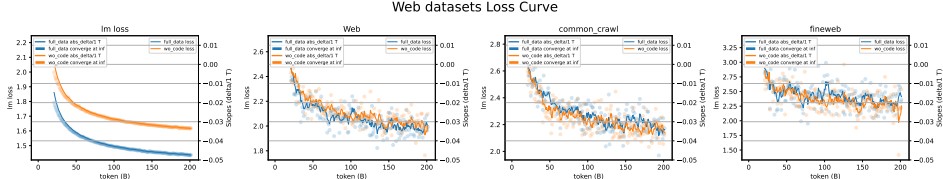

Figure 10: Comparison of loss curves between without code and baseline in the Web domain.

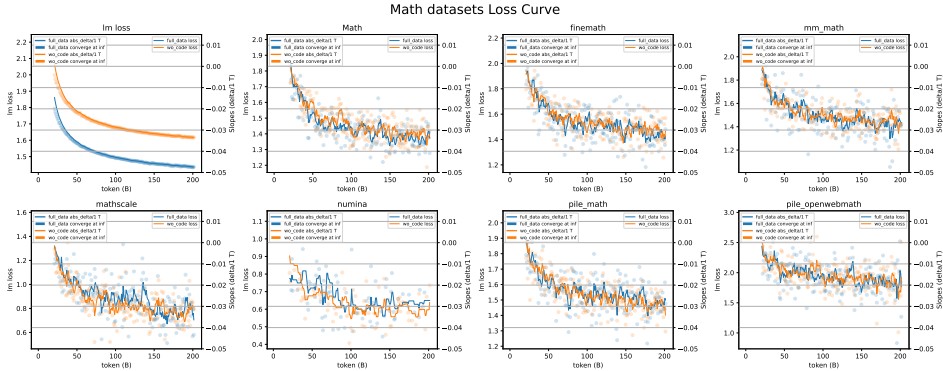

Figure 11: Comparison of loss curves between without code and baseline in the Math domain.

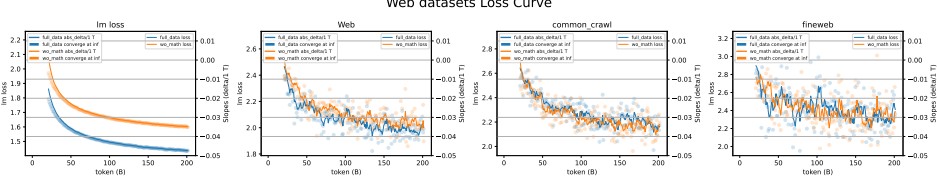

Figure 12: Comparison of loss curves between without math and baseline in the Web domain.

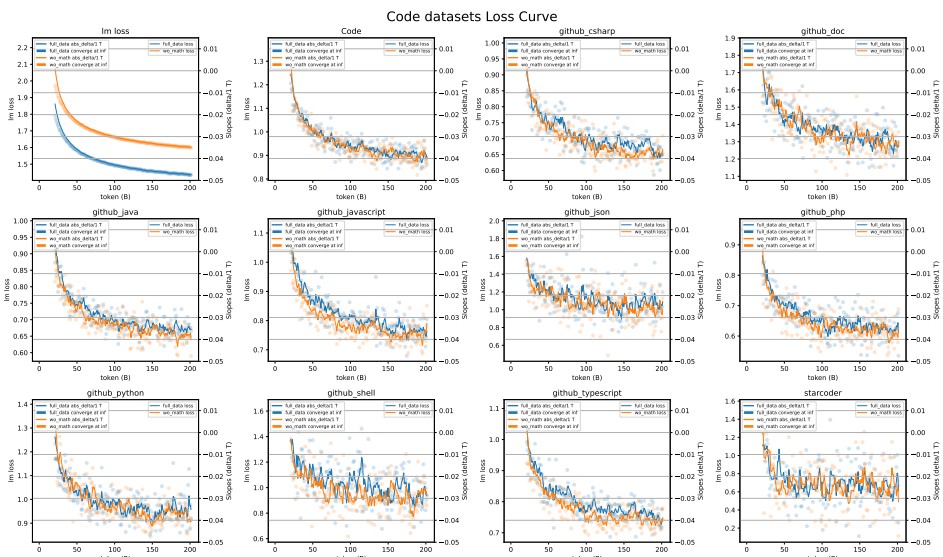

Figure 13: Comparison of loss curves between without code and baseline in the Code domain.

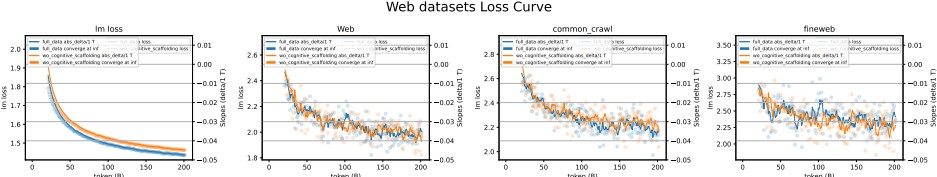

Figure 14: Comparison of loss curves between without cog and baseline in the Web domain.

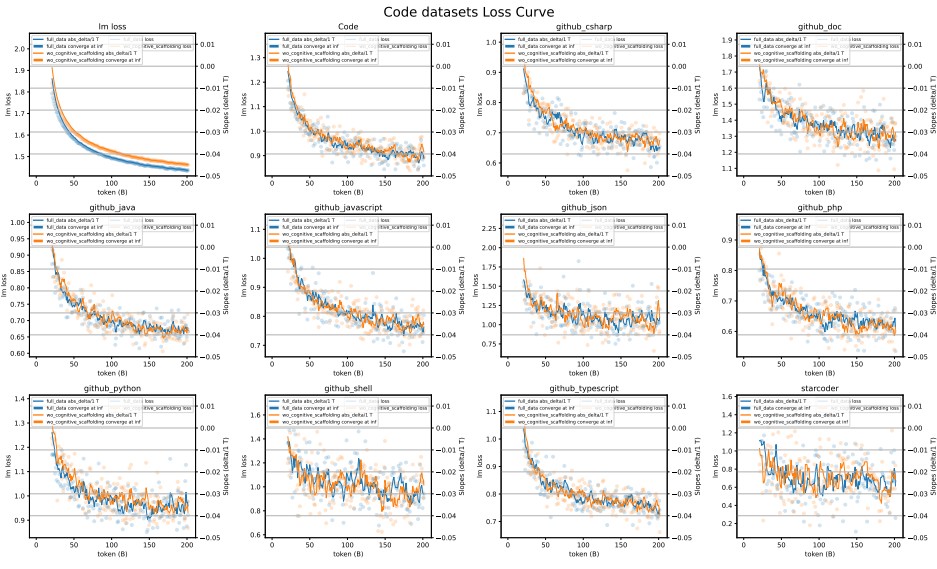

Figure 15: Comparison of loss curves between without cog and baseline in the Code domain.

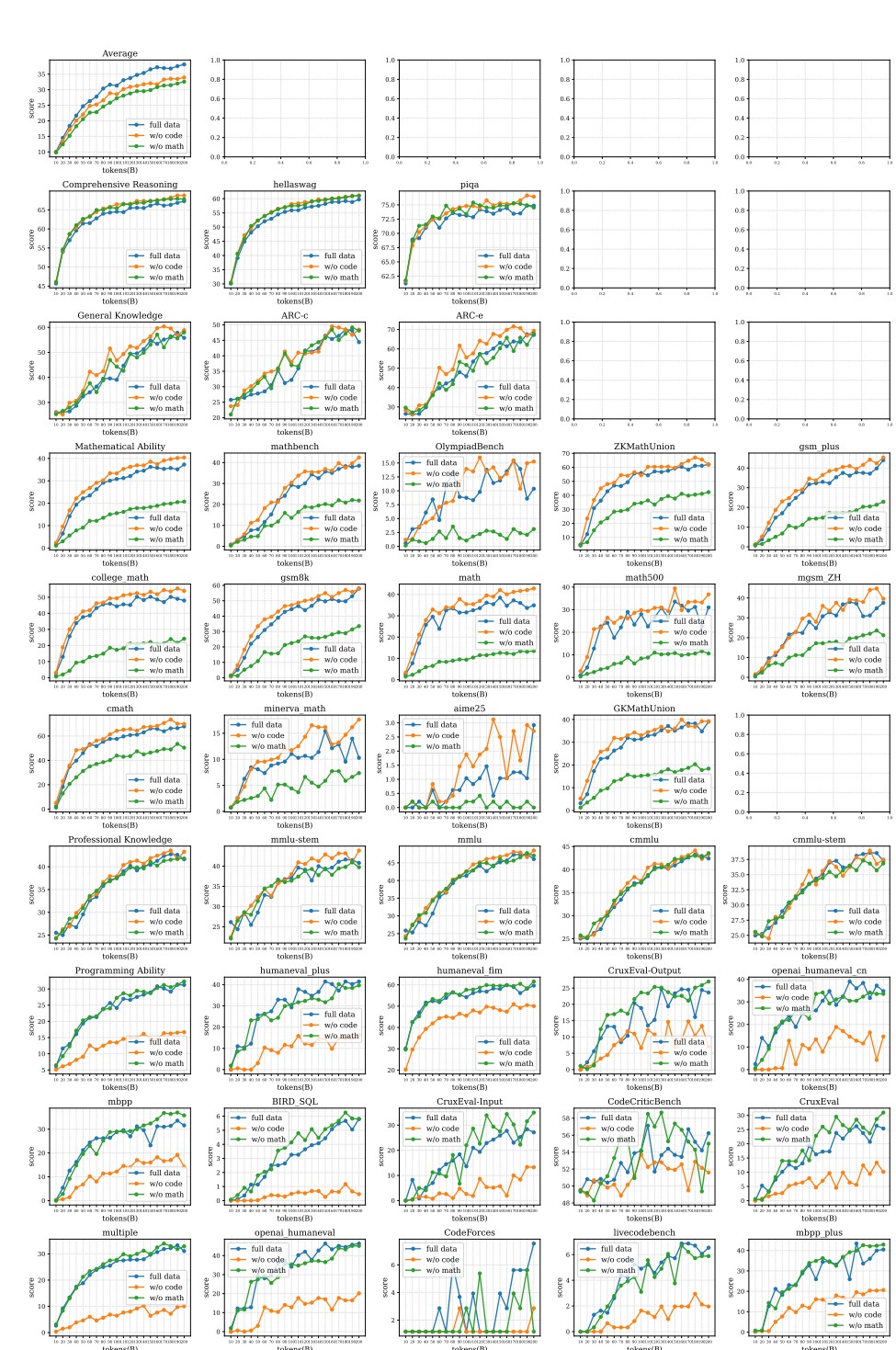

Figure 16: Evaluation results of different data configurations across all benchmarks.

