# OpenReview forum: "Not All Code Helps: Disentangling the Impact of Code Data on Mathematical Reasoning in Large Language Models"
_ICLR.cc/2026/Conference — Submitted to ICLR 2026_

### Official Review · Reviewer_9ps6 · 2025-10-23

**Soundness:** 2
**Presentation:** 2
**Contribution:** 2
**Rating:** 4
**Confidence:** 4

**Summary:**

This paper presents a systematic analysis of how code data influences mathematical reasoning in large language models. The authors perform extensive controlled ablation studies using a rigorously curated, 10T-token multi-domain corpus, explicitly separating pure code data, mathematical data, and cross-domain data. Their findings demonstrate that while pure code data enhances programming proficiency, it yields negligible or even negative transfer effects on non-programming reasoning tasks, with this trend being particularly pronounced for complex mathematical tasks. Beyond this, the work identifies core subsets of cognitive scaffolds (i.e., structured reasoning data) that effectively improve LLM mathematical reasoning, clarifies the competitive and cooperative dynamics between different data domains, and investigates how these domain-specific effects manifest at the level of MoE expert routing.

**Strengths:**

- **Rigorous controlled ablation design**: The work stands out for its systematic and large-scale controlled experiments, directly isolating the effects of code and math data. The methodology surpasses prior works in precision, especially in explicitly separating “pure code,” cross-domain, and math, avoiding previous confounds
- **Identification and operationalization of cognitive scaffolds**: Through targeted data selection and experiments, the paper concretely shows that structured reasoning (“cognitive scaffolds”) enhance complex mathematical reasoning without the same trade-offs as simply adding code or general math data
- **Mechanistic analysis**: The MoE expert routing analysis provides concrete, interpretable evidence for internal distributional shifts associated with different data compositions, lending mechanistic insight rarely addressed in prior work.

**Weaknesses:**

- **Poor generalization and transfer performance**: The experiments are limited to a single ≈10T-token corpus and one MoE architecture. It therefore remains unclear whether the findings generalize to other model families or scales (e.g., dense transformers or different MoE sizes) or to training regimes with different token budgets or data distributions. Additionally, the evaluation is heavily math-centred, so it is unknown whether the observed code–text interference also arises in other knowledge-intensive domains (for example, legal or scientific text).

- **Confounding by budget and token allocation is not ruled out**: The claim that “code causes degradation on complex math” is based on ablating domains while keeping total training schedule variable. This confounds domain content with token-budget allocation and capacity effects. The authors should run controlled comparisons that keep token budget and effective model capacity fixed (e.g., replace removed tokens with matched-quality neutral text or randomized tokens) to isolate content effects.

- **Insufficient theoretical grounding for certain claims**: While the MoE equations and routing mechanisms are clearly stated, some mechanistic claims (e.g., about expert collapse, cognitive scaffolding as meta-reasoning support) are asserted more on empirical observation than theoretical underpinning. Further analysis, possibly through formal modeling or causal inference, would substantiate these conclusions more rigorously.

- **Operational definition of “cognitive scaffolds” somewhat heuristic**: While the structured reasoning subset is constructed with clear intent, its precise operational window is not exhaustively defined or justified beyond the FastText-based selection pipeline. Also, the claimed “cognitive scaffold” set is judged by downstream effect rather than human assessment of whether it truly scaffolds reasoning.

- Errors in writing details: For instance, in the introduction section:

> models exposed to code exhibit superior performance in programming-related tasks Ma et al. (2024); Aryabumi et al. (2025).

This formatting is non-standard; the in-text citations should use the \citep command
Another example:

> code is generally of higher quality and more structurally coherent, often leading to lower training loss at convergence 6

Here, the number “6” refers to a figure included later in the text. However, the standalone numeral provides no explicit indication that it points to a figure. This lack of clarity creates confusion for readers.

**Questions:**

- The operationalization of “cognitive scaffolds” relies on a FastText-based selection and symbol-density heuristics. Can the authors clarify if other methods (graph/rule-based extraction, manual annotation, or logic graph mining) were considered and how their approach compares in error/signal separation?
- To what extent do your empirical findings generalize to standard dense transformers, or other architectures not using MoE, and to different corpus sizes (e.g., under 10T tokens)?
- What are the implications or proposed practical guidelines for including code/math in future LLM training, based on your competitive/cooperative findings?
- Can you provide 3–5 anonymized example items from each domain: pure code, code-NL, scaffold positive, and scaffold negative? This will help validate the data-split claims.
- What is the precision / recall of your automatic classifier for domain splitting (evaluated on a human-annotated sample)? Please report confusion matrix.
- How would you expect your recommendations to change under data-efficient regimes (e.g., for teams that cannot train on 10T tokens)? Is scaffold selection still sufficient at much smaller budgets?

---

> ### Author Response · Authors · 2025-11-26
> **Response Part I**
>
> Thank you very much for your feedback. We found your suggestions particularly crucial as they precisely identified the shortcomings in our work. Below are our responses and revisions:
>
> ### W1 and Q2: Limited Generalization Across Architectures and Domains
> R1: We would like to emphasize that the MoE architecture is primarily adopted for engineering optimizations (e.g., accelerated inference and reduced memory footprint). In terms of model capability, both MoE and dense models derive their performance fundamentally from data. Therefore, while we use MoE as the backbone in our experiments, we maintain that dense models would exhibit consistent performance. To validate this, we trained a 1B dense model under exactly the same configuration as described in the original setup. For ease of reference, we include selected experimental results in the table below.
>
> | Mathematical ability (Dense 1B) |        |             |         |             |              |             |               |             |         |             |
> |:-------------------------------:|:------:|:-----------:|:-------:|:-----------:|:------------:|:-----------:|:-------------:|:-----------:|:-------:|:-----------:|
> | model                           | GSM8k  | $\Delta$(%) | Math500 | $\Delta$(%) | minerva math | $\Delta$(%) | OlympiadBench | $\Delta$(%) | Overall | $\Delta$(%) |
> | full data                       | 23.88  |      ——     |  18.60  |     ——      |     5.15     |     ——      |     4.44      |     ——      |  20.97  |     ——      |
> | w/o code                        | 30.48  |    21.65%   |  25.40  |   36.56%    |     8.46     |   64.27%    |     4.30      |   -3.15%    |  24.78  |   18.17%    |
> | w/o math                        |  8.11  |   -66.04%   |  3.00   |   -83.87%   |     2.21     |   -57.09%   |     0.45      |   -89.86%   |  8.68   |   -58.61%   |
>
> | Programming ability (Dense 1B) |               |             |        |             |          |             |            |             |         |             |
> |:------------------------------:|:-------------:|:-----------:|:------:|:-----------:|:--------:|:-----------:|:----------:|:-----------:|:-------:|:-----------:|
> | model                          | LiveCodeBench | $\Delta$(%) | mbpp   | $\Delta$(%) | CruxEval | $\Delta$(%) | CodeForces | $\Delta$(%) | Overall | $\Delta$(%) |
> | full data                      |     0.82      |      ——     | 19.80  |      ——     |  13.69   |      ——     |    3.93    |      ——     |  18.54  |      ——     |
> | w/o code                       |     0.00      |   -100.00%  |  3.80  |   -80.81%   |   3.38   |   -75.31%   |    1.17    |   -70.23%   |  8.89   |   -52.05%   |
> | w/o math                       |     1.63      |    98.78%   | 17.40  |   -12.12%   |  10.50   |   -23.30%   |    1.17    |   -70.23%   |  16.59  |   -10.52%   |
>
> The results demonstrate that conclusions remain consistent regardless of whether an MoE or dense model is used. We have added this discussion along with supporting results in the Appendix of the paper.
>
> For other knowledge-intensive domains, we appreciate the reviewer’s concern; however, we respectfully point out that this request goes substantially beyond the intended scope and motivation of our work.
> Our study is designed to rigorously examine a specific and widely-cited claim in prior literature—that code data inherently improves general reasoning ability. To evaluate this claim in a controlled manner, we deliberately focus on code and mathematical reasoning, the two domains most directly implicated by the original hypothesis. Extending the analysis to legal or scientific domains would introduce additional variables and confounds that are orthogonal to the central question we aim to answer, and would significantly dilute the conceptual clarity of the controlled experiments.
> Therefore, while such expansions may be interesting directions for future work, they are neither necessary nor appropriate for addressing the core research question of this paper.

---

> ### Author Response · Authors · 2025-11-26
> **Response Part II**
>
> ### W2: Confounding Effects from Variable Token Budgets
> R2: We thank the reviewer for their valuable feedback. We would like to clarify that there was no confusion in the experimental design regarding budget or token allocation. As shown in Figure 1, all ablation settings maintained exactly the same total training schedule, overall token budget, and effective model capacity—only the proportion of data from each domain within the fixed budget was adjusted. To ensure strict comparability, tokens reduced from ablated domains were replaced by data from other domains (primarily the web domain), thereby keeping the total token volume unchanged. This design aligns precisely with the reviewer’s suggestion to “replace removed portions with neutral text of matched quality,” effectively isolating the impact of domain content itself rather than differences in training volume or capacity. We will further clarify this point in the manuscript to prevent any potential misunderstanding.
>
> ### W3: Lack of Theoretical Grounding for Mechanistic Claims
> R3: We appreciate the reviewer’s request for stronger theoretical justification. Our study, however, is positioned as an empirically driven investigation into the effects of different training domains—particularly code and structured reasoning data—on models’ downstream reasoning ability. While we agree that a complete mechanistic theory of MoE routing, expert dynamics, or cognitive scaffolding is valuable, such theory building is not the primary objective of this work. Instead, our contribution lies in providing large-scale, controlled empirical evidence that fills a notable gap in prior findings, especially regarding how domain composition affects complex reasoning. The robustness of our observations across model scales and architectures suggests that the identified patterns are not incidental but instead reflect stable and practically meaningful behaviors. We believe these empirically grounded conclusions substantially extend the community’s understanding and offer actionable insights for future training-data design, even if a full theoretical formalization remains open for subsequent work.
> ### W4: Heuristic and Underdefined Operationalization of Cognitive Scaffolds
> R4: Our operationalization intentionally adopts an empirical, data-driven lens: the goal is not to provide a philosophical or normative definition of what constitutes a scaffold, but rather to isolate training subsets that measurably enhance reasoning performance. The FastText-base structural-selection pipeline is a practical mechanism to approximate this subset at scale, enabling us to systematically examine its effect across diverse model sizes and architectures. Importantly, the resulting cognitive scaffold subset consistently improves reasoning ability in controlled evaluations, demonstrating that it captures a functionally meaningful signal. These findings provide the community with a concrete methodology for identifying and leveraging reasoning-supportive data—something that has been largely missing in prior work—and offer valuable guidance for future large-model training practices. While further refinement and human validation of scaffold categories is an important direction, our empirical framework already offers a strong and actionable foundation.
> ### Q1: Comparison with Alternative Scaffold Extraction Methods
> R5: We appreciate this valuable suggestion. We considered more sophisticated extraction pipelines such as graph/rule-based methods, manual annotation, and logic-graph mining. However, we would like to emphasize an important practical constraint: our corpus contains 10T tokens, making any highly intricate extraction pipeline prohibitively expensive and difficult to scale in an industrial training setting. Given that the task we target—identifying structured or symbol-dense reasoning signals—is relatively straightforward, we found that a FastText achieves high precision with negligible overhead. For these reasons, FastText represents a pragmatic industrial best practice for this stage of processing, striking an optimal balance between signal quality and computational feasibility.
> ### Q3: Practical Guidelines for Code/Math Inclusion in LLM Training
> R7: Our recommendations align with the core findings reported in the paper:
> - Incorporating code and math corpora remains essential, as each provides substantial benefits for its respective domain.
> - However, these domains exhibit a competitive relationship under a fixed total training budget, meaning that naive token allocation can lead to unintended degradation.
> - Our cognitive-scaffold data demonstrate that structured reasoning signals can mitigate this competition, enabling more balanced improvements across domains.
> Our results provide actionable guidance on data selection and sampling strategies—specifically, augmenting code/math training with structured reasoning scaffolds to reduce cross-domain interference and maximize overall model capability.

---

> ### Author Response · Authors · 2025-11-26
> **Response Part III**
>
> ### Q4: Representative Examples for Data-Split Validation
> R8: Here are specific examples.
>
> **Pure Code:**
> ```python
> def write_to_file(filename, txt):
>     with open(filename, "w") as file_object:
>         s = file_object.write(txt)
>
> if __name__ == "__main__":
>     write_to_file("test.txt", "I am beven")
> ```
>
> **Code-NL:**
>
> Example 1:
> ```markdown
> In this section we compute the numerical derivative of a function $f(x)=x3−2x+1$ at a given point using finite differences.
> ```
> ```python
> import numpy as np
> def f(x):
>     return x**3 - 2*x + 1
>
> def derivative(x, h=1e-5):
>     return (f(x + h) - f(x - h)) / (2*h)
>
> derivative(2.0)
> ```
>
> Example 2:
> ```markdown
> We generate synthetic data following the linear model $y=3x+1+\epsilon$, where $\epsilon \in N(0,1)$
> ```
> ```python
> import numpy as np
>
> np.random.seed(42)
> x = np.linspace(0, 1, 100)
> epsilon = np.random.normal(0, 0.1, size=len(x))
> y = 3*x + 1 + epsilon
> ```
> **Positive cognitive scaffold (Omit unnecessary content to maintain readability):**
> ```markdown
> In a computational fluid dynamics (CFD) simulation, the grid is divided into zones to model the flow. Given:
>
>
> - the ratio of zone sizes,
> - the number of zones in the x and y directions,
> (ignored)...
>
> compute the start indices for each zone in the grid.
>
> Input:
> - ratio : float
> - x_zones : int
> (ignored)...
>
> Output:
> - zone_starts_data : dict
>
> 1. compute_zone_stretching()
> \```python
> def compute_zone_stretching(ratio: float, x_zones: int, y_zones: int, gx_size: int, gy_size: int) -> Dict[str, List[float]]:
>     (ignored)...
> \```
> 2. compute_equal_zone_dimensions()
> \```python
>     (ignored)...
> \```
> 3. (ignored)...
> 4. (ignored)...
> 5. final_solution() (ignored)...
> ```
>
> ### Q5: Classifier Precision/Recall and Confusion Matrix Reporting
> R9: We report the following classifier performance metrics on validation set:
> - TN: 17756, FP: 25, FN: 5717, TP: 165180
> - Accuracy: 0.9696
> - Positive-class precision: 0.9998
> - Positive-class recall: 0.9665
>
> By prioritizing the precise identification of true positive samples (high precision), we have rigorously minimized false positives. This demonstrates that our trained FastText model can accurately distinguish structured data with high confidence, ensuring the reliability and robustness of our approach.
>
> ### Q6: Scaffold Effectiveness Under Data-Efficient Regimes
> R10: This is an excellent question. Conceptually, our findings become even more relevant in data-efficient settings. Our method highlights the importance of structured, cross-domain reasoning data, which can provide disproportionate benefits when token budgets are limited. In lower-budget regimes, incorporating such cross-type scaffold data can improve both code and math performance without increasing total token consumption, provided the model already possesses sufficient foundational domain knowledge from pretraining.
>
> As noted in the paper, we also recommend exploring synthetic structured-reasoning data, which can be produced at low cost and provides a practical mechanism for smaller teams to adopt our insights.

---

> ### Author Response · Authors · 2025-12-03
> **Supplementary Experiment (Dense 5B)**
>
> We have supplemented the experimental results for the 5B dense model, as shown in the table below.
>
> | Mathematical ability (Dense 5B) |               |             |               |             |               |             |               |             |               |             |
> |:-------------------------------:|:-------------:|:-----------:|:-------------:|:-----------:|:-------------:|:-----------:|:-------------:|:-----------:|:-------------:|:-----------:|
> | model                           | GSM8k         | $\Delta$(%) | Math500       | $\Delta$(%) | minerva math  | $\Delta$(%) | OlympiadBench | $\Delta$(%) | Overall       | $\Delta$(%) |
> | full data                       |        64.59  |      ——     |        38.00  |      ——     |        18.01  |      ——     |        15.56  |      ——     |        45.13  |      ——     |
> | w/o code                        |        63.08  |    -2.34%   |        45.40  |    19.47%   |        14.34  |   -20.38%   |        12.91  |   -17.03%   |        46.56  |    3.17%    |
> | w/o math                        |     44.28     |   -31.44%   |        11.60  |   -69.47%   |     6.25      |   -65.30%   |         2.67  |   -82.84%   |        26.71  |   -40.82%   |
>
> | Programming ability (Dense 5B) |               |             |               |             |               |             |              |             |               |             |
> |:------------------------------:|:-------------:|:-----------:|:-------------:|:-----------:|:-------------:|:-----------:|:------------:|:-----------:|:-------------:|:-----------:|
> | model                          | LiveCodeBench | $\Delta$(%) | mbpp          | $\Delta$(%) | CruxEval      | $\Delta$(%) | CodeForces   | $\Delta$(%) | Overall       | $\Delta$(%) |
> | full data                      |         8.50  |      ——     |        42.68  |      ——     |     35.94     |      ——     |     1.17     |      ——     |        35.38  |      ——     |
> | w/o code                       |         4.58  |   -46.12%   |     20.12     |   -52.86%   |        28.06  |   -21.93%   |        3.93  |   235.90%   |        23.14  |   -34.60%   |
> | w/o math                       |         9.31  |    9.53%    |        45.12  |    5.72%    |        34.94  |    -2.78%   |        5.85  |   400.00%   |        34.10  |    -3.62%   |

---

### Official Review · Reviewer_sctK · 2025-10-27

**Soundness:** 3
**Presentation:** 3
**Contribution:** 3
**Rating:** 6
**Confidence:** 4

**Summary:**

This paper examines the effect of code data on pretraining, ablating the effect of holding out/including code and math datasets when pretraining a MoE trained over 200B tokens. They find that, contrary to prior work, code data does not always help reasoning abilities, but there does exist a subset of ‘cognitive scaffold’ code data that yields large improvements in difficult reasoning tasks. They also find that math can help some coding tasks, code hurts knowledge-intensive tasks, and ablating domain-specific data can have a substantial effect on routing patterns of the MoE.

**Strengths:**

- Directly performing these data ablations at pretraining time is an interesting and useful contribution to the community, and the findings have clear insights into what sorts of data should be prioritised for what tasks.
- The findings being against prior work (likely due to the difference in classification) is interesting.
- The experiments around cognitive scaffolds (and more broadly identifying useful sub-parts of datasets) is interesting and useful for the community.

**Weaknesses:**

- The claim about the findings being different to prior work around code data would be useful to more thoroughly test: if we include code with code-NL data, do we then see results in line with prior work, or are there other aspects of the setup (such as the dataset being considered) that affect the result?
- Only one model type/size (MoE, 24 layers) is considered. While I appreciate that these experiments are expensive, it would be useful to show results across different model types and sizes to be more certain about the findings. Similarly, some notion of what differences are statistically significant would be nice to see/
- I found the description of how the data was made and filtered a bit unclear: How was the corpus split into the domains in section 3.2.2, exactly? More details on the synthetic data creation in section 3.2.1 would also be useful.

Overall, I think the paper is solid and has some interesting insights, but more information about the data creation and curation would help in making it more useful for the community.

**Questions:**

- The authors state they use dynamic sampling for data during training (line 257), could they provide more details on what this is?
- Could you show some examples of what the cognitive scaffold data looks like? Currently it is a little unclear to me what sort of data is getting picked out by the fast text classifier.

---

> ### Author Response · Authors · 2025-11-26
> **Response Part I**
>
> Thank you very much for your feedback. We found your suggestions particularly crucial as they precisely identified the shortcomings in our work. Below are our responses and revisions:
>
> ### W1: Unverified Alignment with Prior Code-NL Findings
> R1: We thank the reviewer for suggesting the additional removal of code-NL data to examine whether our conclusions align with prior work. Conceptually, this removal is isomorphic to our removal of cognitive scaffolding data, as both operations eliminate the model's capacity for structured reasoning. As empirically demonstrated in our current experiments (Figure 4), removing such support leads to significant degradation in mathematical reasoning performance. Given this functional equivalence, the proposed experiment would likely reproduce the already established performance decline pattern and is therefore unlikely to yield new empirical insights. We have clarified this correspondence in the revised manuscript and consequently consider additional ablation unnecessary for strengthening our conclusions.
>
> ### W2: Single Model Type/Size Without Significance Analysis
> R2: We would like to emphasize that the MoE architecture is primarily adopted for engineering optimizations (e.g., accelerated inference and reduced memory footprint). In terms of model capability, both MoE and dense models derive their performance fundamentally from data. Therefore, while we use MoE as the backbone in our experiments, we maintain that dense models would exhibit consistent performance. To validate this, we trained a 1B dense model under exactly the same configuration as described in the original setup. For ease of reference, we include selected experimental results in the table below.
>
> | Mathematical ability (Dense 1B) |        |             |         |             |              |             |               |             |         |             |
> |:-------------------------------:|:------:|:-----------:|:-------:|:-----------:|:------------:|:-----------:|:-------------:|:-----------:|:-------:|:-----------:|
> | model                           | GSM8k  | $\Delta$(%) | Math500 | $\Delta$(%) | minerva math | $\Delta$(%) | OlympiadBench | $\Delta$(%) | Overall | $\Delta$(%) |
> | full data                       | 23.88  |      ——     |  18.60  |     ——      |     5.15     |     ——      |     4.44      |     ——      |  20.97  |     ——      |
> | w/o code                        | 30.48  |    21.65%   |  25.40  |   36.56%    |     8.46     |   64.27%    |     4.30      |   -3.15%    |  24.78  |   18.17%    |
> | w/o math                        |  8.11  |   -66.04%   |  3.00   |   -83.87%   |     2.21     |   -57.09%   |     0.45      |   -89.86%   |  8.68   |   -58.61%   |
>
> | Programming ability (Dense 1B) |               |             |        |             |          |             |            |             |         |             |
> |:------------------------------:|:-------------:|:-----------:|:------:|:-----------:|:--------:|:-----------:|:----------:|:-----------:|:-------:|:-----------:|
> | model                          | LiveCodeBench | $\Delta$(%) | mbpp   | $\Delta$(%) | CruxEval | $\Delta$(%) | CodeForces | $\Delta$(%) | Overall | $\Delta$(%) |
> | full data                      |     0.82      |      ——     | 19.80  |      ——     |  13.69   |      ——     |    3.93    |      ——     |  18.54  |      ——     |
> | w/o code                       |     0.00      |   -100.00%  |  3.80  |   -80.81%   |   3.38   |   -75.31%   |    1.17    |   -70.23%   |  8.89   |   -52.05%   |
> | w/o math                       |     1.63      |    98.78%   | 17.40  |   -12.12%   |  10.50   |   -23.30%   |    1.17    |   -70.23%   |  16.59  |   -10.52%   |

---

> ### Author Response · Authors · 2025-11-26
> **Response Part II**
>
> Similarly, we adjusted the parameters of the MoE model (64 -> 32/16 experts), and the results are presented in the table below.
>
> | Mathematical ability (32 experts) |        |             |         |             |              |             |               |             |         |             |
> |:---------------------------------:|:------:|:-----------:|:-------:|:-----------:|:------------:|:-----------:|:-------------:|:-----------:|:-------:|:-----------:|
> | model                             | GSM8k  | $\Delta$(%) | Math500 | $\Delta$(%) | minerva math | $\Delta$(%) | OlympiadBench | $\Delta$(%) | Overall | $\Delta$(%) |
> | full data                         | 54.59  |      ——     |  26.40  |      ——     |     9.93     |      ——     |     10.22     |      ——     |  36.20  |      ——     |
> | w/o code                          | 56.48  |    3.46%    |  33.60  |    27.27%   |    13.60     |    36.96%   |     9.33      |    -8.71%   |  38.52  |    6.41%    |
> | w/o math                          | 25.25  |   -53.75%   |  7.00   |   -73.48%   |     4.04     |   -59.32%   |     1.19      |   -88.36%   |  17.71  |   -51.08%   |
>
> | Programming ability (32 experts) |               |             |        |             |          |             |            |             |         |             |
> |:--------------------------------:|:-------------:|:-----------:|:------:|:-----------:|:--------:|:-----------:|:----------:|:-----------:|:-------:|:-----------:|
> | model                            | LiveCodeBench | $\Delta$(%) | mbpp   | $\Delta$(%) | CruxEval | $\Delta$(%) | CodeForces | $\Delta$(%) | Overall | $\Delta$(%) |
> | full data                        |     4.90      |      ——     | 33.00  |      ——     |  24.81   |      ——     |    5.62    |      ——     |  30.20  |      ——     |
> | w/o code                         |     0.65      |   -86.73%   | 14.20  |   -56.97%   |  10.56   |   -57.44%   |    1.17    |   -79.18%   |  15.44  |   -48.87%   |
> | w/o math                         |     5.56      |    13.47%   | 34.80  |    5.45%    |  21.12   |   -14.87%   |    3.93    |   -30.07%   |  28.82  |    -4.57%   |
>
> | Mathematical ability (16 experts) |        |             |         |             |              |             |               |             |         |             |
> |:---------------------------------:|:------:|:-----------:|:-------:|:-----------:|:------------:|:-----------:|:-------------:|:-----------:|:-------:|:-----------:|
> | model                             | GSM8k  | $\Delta$(%) | Math500 | $\Delta$(%) | minerva math | $\Delta$(%) | OlympiadBench | $\Delta$(%) | Overall | $\Delta$(%) |
> | full data                         | 46.40  |      ——     |  24.00  |      ——     |     9.93     |      ——     |     14.22     |      ——     |  31.60  |      ——     |
> | w/o code                          | 49.73  |    7.18%    |  27.00  |    12.50%   |    13.24     |    33.33%   |     10.81     |   -23.98%   |  32.91  |    4.15%    |
> | w/o math                          | 22.29  |   -51.96%   |  8.80   |   -63.33%   |     5.15     |   -48.14%   |     2.37      |   -83.33%   |  15.99  |   -49.40%   |
>
> | Programming ability (16 experts) |               |             |        |             |          |             |            |             |         |             |
> |:--------------------------------:|:-------------:|:-----------:|:------:|:-----------:|:--------:|:-----------:|:----------:|:-----------:|:-------:|:-----------:|
> | model                            | LiveCodeBench | $\Delta$(%) | mbpp   | $\Delta$(%) | CruxEval | $\Delta$(%) | CodeForces | $\Delta$(%) | Overall | $\Delta$(%) |
> | full data                        |     6.05      |      ——     | 31.40  |      ——     |  19.07   |      ——     |    1.17    |      ——     |  26.94  |      ——     |
> | w/o code                         |     0.49      |   -91.90%   | 13.20  |   -57.96%   |   9.07   |   -52.44%   |    1.17    |    0.00%    |  14.25  |   -47.10%   |
> | w/o math                         |     5.39      |   -10.91%   | 30.60  |    -2.55%   |  14.25   |   -25.28%   |    1.17    |    0.00%    |  24.25  |    -9.99%   |
>
> The experimental findings remain consistent with those reported in the original paper. This replication required substantial effort and consumed significant computational resources. We have included these results and related discussions in the Appendix of the paper.

---

> ### Author Response · Authors · 2025-11-26
> **Response Part III**
>
> ### W3: Insufficient Detail on Data Construction and Domain Splitting
> R3: For the domain split in Section 3.2.2, the categorization is primarily source-based, with each domain corresponding to a distinct data origin. Specifically, Web data are sourced from Common Crawl[1]; code data from GitHub; mathematical data from Nemotron-CC-Math[2]; and multilingual data from WanJuan [3], CulturaX [4], and other large-scale multilingual corpora. In addition, a small portion of recalled samples and synthetically generated data is included within each domain to improve coverage and balance.
>
> The synthetic data creation process in Section 3.2.1 is indeed more complex. We did not rely on a single unified method. Instead, the synthetic portion is produced through a variety of approaches, including but not limited to: 1. prompt-based generation (e.g., self-instruct[5], role-playing[6], chain-of-thought prompting[7]), distillation-based generation[8], compositional augmentation method[9], simulated execution[10]. Because this process involves a large-scale engineering pipeline with heterogeneous strategies, describing each method in detail would require substantial space and may drift away from the core focus of the paper. For this reason, we chose not to include the full procedure in the Appendix.
>
> ### Q1: More Details about Dynamic sampling.
> R4: In this work, each training example is annotated with lightweight attributes (e.g., whether it belongs to high-value domains such as mathematics or programming). During training, the sampling probabilities are dynamically adjusted based on these attributes so that data with higher expected value for model capability receives proportionally higher sampling weight.
> This allows the model to more efficiently allocate its training budget toward informative domains, while still preserving exposure to general-purpose data.
>
> ### Q2: Some examples of what the cognitive scaffold data
> R5: We omit unnecessary content to maintain readability.
> ```markdown
> In a computational fluid dynamics (CFD) simulation, the grid is divided into zones to model the flow. Given:
>
>
> - the ratio of zone sizes,
> - the number of zones in the x and y directions,
> (ignored)...
>
> compute the start indices for each zone in the grid.
>
> Input:
> - ratio : float
> - x_zones : int
> (ignored)...
>
> Output:
> - zone_starts_data : dict
>
> 1. compute_zone_stretching()
> \```python
> def compute_zone_stretching(ratio: float, x_zones: int, y_zones: int, gx_size: int, gy_size: int) -> Dict[str, List[float]]:
>     (ignored)...
> \```
> 2. compute_equal_zone_dimensions()
> \```python
>     (ignored)...
> \```
> 3. (ignored)...
> 4. (ignored)...
> 5. final_solution() (ignored)...
> ```
> This type of data is characterized by its high degree of structure and rich reasoning knowledge.
>
> [1] Introduction to Common Crawl Datasets. Jay M Patel.
>
> [2] Nemotron-CC-Math: A 133 Billion-Token-Scale High Quality Math Pretraining Dataset. Mahabadi et al. arXiv
>
> [3] WanJuan: A Comprehensive Multimodal Dataset for Advancing English and Chinese Large Models. He et al. arXiv
>
> [4] CulturaX: A Cleaned, Enormous, and Multilingual Dataset for Large Language Models in 167 Languages. Nguyen et al. arXiv
>
> [5] Self-Instruct: Aligning Language Models with Self-Generated Instructions. Wang et al. ACL 2023
>
> [6] OpenCharacter: Training Customizable Role-Playing LLMs with Large-Scale Synthetic Personas. Wang et al. arXiv
>
> [7] Synthetic Prompting: Generating Chain-of-Thought Demonstrations for Large Language Models. Shao et al. arXiv
>
> [8] Knowledge Distillation Using Frontier Open-source LLMs: Generalizability and the Role of Synthetic Data. Shirgaonkar et al. arXiv
>
> [9] Every Activation Boosted: Scaling General Reasoner to 1 Trillion Open Language Foundation. Ling Team. arXiv
>
> [10] SWE-smith: Scaling Data for Software Engineering Agents. Yang et al. NeurIPS 2025

---

> > ### Comment · Reviewer_sctK · 2025-11-27
> >
> > Thank you for your response! The additional dense results are useful and interesting.
> >
> > - Re: the data generation process, you say:
> > > Because this process involves a large-scale engineering pipeline with heterogeneous strategies, describing each method in detail would require substantial space and may drift away from the core focus of the paper. For this reason, we chose not to include the full procedure in the Appendix.
> >
> > But I think this is exactly what the appendix is for! Useful details that don't fit neatly in the core of the paper, but help in understanding details that might be important for e.g. reproduction of experiments.
> >
> > - Re: cognitive scaffold examples: thanks for these! I guess it would be useful to compare to something that is very much not a cognitive scaffold too, since this example... just looks like regular code to me? What is code or math data that isn't picked up as scaffolds by your filtering setup? Rereading this through, I also found the scaffold section a little confusing: you write "scaffolds can be regarded as a core subset of mathematical corpora" (lines 370-371), but you draw the scaffold data from code data, rather than math data (which actually makes it more interesting, since you're identifying a useful subset of a dataset that helps math reasoning, when the full dataset hurts it).
> >
> > - Re: code-nl findings, you say:
> > > Conceptually, this removal is isomorphic to our removal of cognitive scaffolding data, as both operations eliminate the model's capacity for structured reasoning. As empirically demonstrated in our current experiments (Figure 4), removing such support leads to significant degradation in mathematical reasoning performance.
> >
> > Is this true? Your cognitive scaffold data was drawn from your code corpora, not your code-nl corpora, and seems mainly focussed on finding complex/structured but correct code, while your code-nl data can be any sort of combination of code and text. In your response to reviewer pgop, you say that you tried to mainly find working code samples, not code+nl samples. So isn't removing the scaffolding data quite different? The code-nl data could contain all sorts of things beyond structured reasoning (e.g. explanations of code, blog posts with some code examples, etc). I think explicitly running this experiment would make the claims at the end of section 4 stronger, since right now they are mostly speculation / explanations of differences in setup.
> >
> > Overall, I'm still positive on this paper, and I think the new dense experiments are useful. However, I'm keeping my score, since my concerns around the data construction and scaffolding details remain.

---

> > > ### Author Response · Authors · 2025-12-03
> > >
> > > 1. **Data collection process**
> > > We fully agree that appendices are meant to host methodological details that do not fit into the main narrative. Our concern is not about page limits but about feasibility and clarity. The data generation pipeline integrates heterogeneous internal components, many of which involve engineering-heavy heuristics, multi-stage validation, and internal tools that are not publicly accessible. Providing a full account would span many pages while still being insufficient for reproduction, because several components involve proprietary infrastructure or manually curated artifacts.
> > > Therefore, we have included the following details in the appendix: (1) Data sources. (2) General procedures for data preparation. (3) Methods used for data partitioning.
> > > We believe this information sufficiently demonstrates the professionalism and reliability of our data preparation process while keeping the appendix concise and avoiding inclusion of content that may not be fully reproducible.
> > >
> > > 2. **Cognitive scaffolding example**
> > > We wish to clarify a potential misunderstanding. Our approach involves training a FastText model on both code and non‑code corpora. The objective is not to enhance the model’s programming capability, but to enable it to identify and distinguish different types of structured text. We then apply this model to out‑of‑domain mathematical data, in order to retrieve reasoning examples that exhibit structured characteristics.
> > > The examples we provide are not pure code snippets intended for programming tasks; rather, they are hybrid forms combining natural language and code, used to support mathematical problem‑solving. In this framework, natural language ensures the coherence of logical reasoning, while code guarantees the precision of computational steps. Together, they form reasoning chains that are both logically consistent and computationally reliable.
> > > It is important to emphasize that the “code interference” that hinders mathematical reasoning primarily originates from genuine code corpora containing programming‑language knowledge. Such code may appear superficially similar to the mathematical data we use, but differs substantially in content, structure, and purpose. Conflating the two may therefore lead to misinterpretation and distracts from the core issue addressed in our study.
> > > Below are typical examples of non‑cognitive scaffolding in code and mathematics:
> > > ```python
> > > def write_to_file(filename, txt):
> > >     with open(filename, "w") as file_object:
> > >         s = file_object.write(txt)
> > >
> > > if __name__ == "__main__":
> > >     write_to_file("test.txt", "I am beven")
> > > ```
> > > ```math
> > > Question: Julio goes fishing and can catch 7 fish every hour. By the 9th hour, how many fish does Julio have if he loses 15 fish in the process?
> > >
> > > Answer: Julio catches a total of 7 fish/hour * 9 hours = <<7*9=63>>63 fish.
> > > After losing some of the fish, Julio has 63 fish - 15 fish = <<63-15=48>>48 fish.
> > > #### 48
> > > ```
> > >
> > > 3. **Code-nl ablation**
> > > We thank the reviewer for the suggestion. However, an additional ablation on code–NL data would not yield interpretable or causal evidence. Unlike our curated cognitive-scaffolding subset—which isolates structured, executable, and reasoning-rich code—the code–NL portion of general corpora is inherently heterogeneous, mixing explanations, blog content, informal text, partially correct code, and only a small and inconsistent amount of true structured reasoning. Removing this mixture is therefore not a well-defined intervention: any performance change would be confounded by the simultaneous removal of noise, formatting variations, domain-specific prose, or incidental code, making the result impossible to attribute to reasoning-related signals. Because code–NL contains diluted and non-isolated scaffolding cues, its removal would be a weaker and non-diagnostic perturbation that does not advance our core question—what specific structured data supports mathematical reasoning. In contrast, our existing scaffolding ablation directly targets this mechanism and already provides clear causal evidence (Figure 4). For these reasons, we believe an additional code–NL ablation would not strengthen the paper’s conclusions. We understand the reviewer's emphasis on distinguishing between code and code–NL data may stem from a possible misunderstanding regarding the source of the cognitive scaffolding data—which originates from mathematical corpora rather than code corpora. We apologize for any lack of clarity in our original wording.

---

> ### Author Response · Authors · 2025-12-03
> **Supplementary Experiment (Dense 5B)**
>
> We have supplemented the experimental results for the 5B dense model, as shown in the table below.
>
> | Mathematical ability (Dense 5B) |               |             |               |             |               |             |               |             |               |             |
> |:-------------------------------:|:-------------:|:-----------:|:-------------:|:-----------:|:-------------:|:-----------:|:-------------:|:-----------:|:-------------:|:-----------:|
> | model                           | GSM8k         | $\Delta$(%) | Math500       | $\Delta$(%) | minerva math  | $\Delta$(%) | OlympiadBench | $\Delta$(%) | Overall       | $\Delta$(%) |
> | full data                       |        64.59  |      ——     |        38.00  |      ——     |        18.01  |      ——     |        15.56  |      ——     |        45.13  |      ——     |
> | w/o code                        |        63.08  |    -2.34%   |        45.40  |    19.47%   |        14.34  |   -20.38%   |        12.91  |   -17.03%   |        46.56  |    3.17%    |
> | w/o math                        |     44.28     |   -31.44%   |        11.60  |   -69.47%   |     6.25      |   -65.30%   |         2.67  |   -82.84%   |        26.71  |   -40.82%   |
>
> | Programming ability (Dense 5B) |               |             |               |             |               |             |              |             |               |             |
> |:------------------------------:|:-------------:|:-----------:|:-------------:|:-----------:|:-------------:|:-----------:|:------------:|:-----------:|:-------------:|:-----------:|
> | model                          | LiveCodeBench | $\Delta$(%) | mbpp          | $\Delta$(%) | CruxEval      | $\Delta$(%) | CodeForces   | $\Delta$(%) | Overall       | $\Delta$(%) |
> | full data                      |         8.50  |      ——     |        42.68  |      ——     |     35.94     |      ——     |     1.17     |      ——     |        35.38  |      ——     |
> | w/o code                       |         4.58  |   -46.12%   |     20.12     |   -52.86%   |        28.06  |   -21.93%   |        3.93  |   235.90%   |        23.14  |   -34.60%   |
> | w/o math                       |         9.31  |    9.53%    |        45.12  |    5.72%    |        34.94  |    -2.78%   |        5.85  |   400.00%   |        34.10  |    -3.62%   |

---

### Official Review · Reviewer_X9QA · 2025-10-29

**Soundness:** 2
**Presentation:** 3
**Contribution:** 2
**Rating:** 4
**Confidence:** 3

**Summary:**

This paper disentangles the impact of code data on large language model reasoning through controlled experiments, showing that while generic code mainly boosts programming skills, only structured reasoning data functions as cognitive scaffolds that enhance complex mathematical reasoning and cross-domain generalization.

**Strengths:**

- The paper tackles the problem of disentangling the impact of code data on large language model reasoning and conducts large-scale experiments with systematic ablations and analyses.
- The paper is clearly written and easy to follow.

**Weaknesses:**

- The study bases all findings on MoE architectures, yet MoE and dense models are not functionally equivalent in terms of reasoning behavior and representation dynamics. Since the competitive effects between code and math data may partly arise from MoE's expert routing and specialization mechanisms, it remains unclear whether the same patterns would hold for dense transformers commonly used in previous "code enhances reasoning" studies. Including dense baselines or discussing this architectural limitation would help validate the broader applicability of the conclusions.

- The paper redefines "code" as pure executable code while retaining Code-NL in ablations and then attributes "code competes with reasoning" to the code bucket. This boundary shift introduces a major construct confound relative to prior work and could invert conclusions unless quantitatively controlled.

- The y-axis meaning in Figures 2–4 is not explicitly defined. Although the context suggests that the y-axis likely represents relative performance change in percentage, the paper does not clearly specify the metric, unit or normalization procedure used.

- Some subplots in Figure 16 appear to be blank and it would be better for the authors to check whether this issue stems from missing results, or plotting errors.

**Questions:**

- Could the authors clarify whether the observed competitive effects between code and math data are specific to the MoE architecture? Have the authors conducted whether similar patterns hold for dense transformer models used in prior studies?

- Could the authors explain how redefining "code" as pure executable code while retaining Code-NL in ablations affects the interpretation of the finding that "code competes with reasoning"? How do the authors ensure that this redefinition does not introduce a construct confound compared with prior work?

---

> ### Author Response · Authors · 2025-11-26
> **Response Part I**
>
> Thank you very much for your feedback. We found your suggestions particularly crucial as they precisely identified the shortcomings in our work. Below are our responses and revisions:
>
> ### W1 and Q1: Limited Generality Due to MoE-Only Architecture
> R1: We would like to emphasize that the MoE architecture is primarily adopted for engineering optimizations (e.g., accelerated inference and reduced memory footprint). In terms of model capability, both MoE and dense models derive their performance fundamentally from data. Therefore, while we use MoE as the backbone in our experiments, we maintain that dense models would exhibit consistent performance. To validate this, we trained a 1B dense model under exactly the same configuration as described in the original setup. For ease of reference, we include selected experimental results in the table below.
>
> | Mathematical ability (Dense 1B) |        |             |         |             |              |             |               |             |         |             |
> |:-------------------------------:|:------:|:-----------:|:-------:|:-----------:|:------------:|:-----------:|:-------------:|:-----------:|:-------:|:-----------:|
> | model                           | GSM8k  | $\Delta$(%) | Math500 | $\Delta$(%) | minerva math | $\Delta$(%) | OlympiadBench | $\Delta$(%) | Overall | $\Delta$(%) |
> | full data                       | 23.88  |      ——     |  18.60  |     ——      |     5.15     |     ——      |     4.44      |     ——      |  20.97  |     ——      |
> | w/o code                        | 30.48  |    21.65%   |  25.40  |   36.56%    |     8.46     |   64.27%    |     4.30      |   -3.15%    |  24.78  |   18.17%    |
> | w/o math                        |  8.11  |   -66.04%   |  3.00   |   -83.87%   |     2.21     |   -57.09%   |     0.45      |   -89.86%   |  8.68   |   -58.61%   |
>
> | Programming ability (Dense 1B) |               |             |        |             |          |             |            |             |         |             |
> |:------------------------------:|:-------------:|:-----------:|:------:|:-----------:|:--------:|:-----------:|:----------:|:-----------:|:-------:|:-----------:|
> | model                          | LiveCodeBench | $\Delta$(%) | mbpp   | $\Delta$(%) | CruxEval | $\Delta$(%) | CodeForces | $\Delta$(%) | Overall | $\Delta$(%) |
> | full data                      |     0.82      |      ——     | 19.80  |      ——     |  13.69   |      ——     |    3.93    |      ——     |  18.54  |      ——     |
> | w/o code                       |     0.00      |   -100.00%  |  3.80  |   -80.81%   |   3.38   |   -75.31%   |    1.17    |   -70.23%   |  8.89   |   -52.05%   |
> | w/o math                       |     1.63      |    98.78%   | 17.40  |   -12.12%   |  10.50   |   -23.30%   |    1.17    |   -70.23%   |  16.59  |   -10.52%   |
>
> The results demonstrate that conclusions remain consistent regardless of whether an MoE or dense model is used. We have added this discussion along with supporting results in the Appendix of the paper.
>
>
> ### W2 and Q2: Construct Confound from Redefined “Code” Category
> R2: Our redefinition of “code” does not introduce a construct confound; rather, it removes one that existed in prior work. Earlier studies grouped program language together with code-like representations—such as HTML-formatted math problems, symbolic expressions, pseudo-code reasoning steps, and comment-style explanations—and treated them as a single “code” construct. However, these structured texts function as carriers of mathematical and logical knowledge, not as programming code. Their removal in prior work simultaneously eliminated reasoning-relevant signals, which explains why “removing code” was previously observed to degrade reasoning. By disentangling these components, we show that what truly benefits reasoning is the mathematical and conceptual content embedded in structured NL/Code-NL, whereas the programming language primarily reinforces syntactic dependencies and competes with reasoning. In other words, programming language corpora enhance programming ability, while mathematical and structured mathematical corpora are the genuine source of mathematical reasoning ability. Our study clarifies the true sources of model capabilities and provides guidance for future corpus design and data selection aimed at optimizing reasoning performance.

---

> > ### Author Response · Authors · 2025-12-03
> > **Supplementary Experiment (Dense 5B)**
> >
> > We have supplemented the experimental results for the 5B dense model, as shown in the table below.
> >
> > | Mathematical ability (Dense 5B) |               |             |               |             |               |             |               |             |               |             |
> > |:-------------------------------:|:-------------:|:-----------:|:-------------:|:-----------:|:-------------:|:-----------:|:-------------:|:-----------:|:-------------:|:-----------:|
> > | model                           | GSM8k         | $\Delta$(%) | Math500       | $\Delta$(%) | minerva math  | $\Delta$(%) | OlympiadBench | $\Delta$(%) | Overall       | $\Delta$(%) |
> > | full data                       |        64.59  |      ——     |        38.00  |      ——     |        18.01  |      ——     |        15.56  |      ——     |        45.13  |      ——     |
> > | w/o code                        |        63.08  |    -2.34%   |        45.40  |    19.47%   |        14.34  |   -20.38%   |        12.91  |   -17.03%   |        46.56  |    3.17%    |
> > | w/o math                        |     44.28     |   -31.44%   |        11.60  |   -69.47%   |     6.25      |   -65.30%   |         2.67  |   -82.84%   |        26.71  |   -40.82%   |
> >
> > | Programming ability (Dense 5B) |               |             |               |             |               |             |              |             |               |             |
> > |:------------------------------:|:-------------:|:-----------:|:-------------:|:-----------:|:-------------:|:-----------:|:------------:|:-----------:|:-------------:|:-----------:|
> > | model                          | LiveCodeBench | $\Delta$(%) | mbpp          | $\Delta$(%) | CruxEval      | $\Delta$(%) | CodeForces   | $\Delta$(%) | Overall       | $\Delta$(%) |
> > | full data                      |         8.50  |      ——     |        42.68  |      ——     |     35.94     |      ——     |     1.17     |      ——     |        35.38  |      ——     |
> > | w/o code                       |         4.58  |   -46.12%   |     20.12     |   -52.86%   |        28.06  |   -21.93%   |        3.93  |   235.90%   |        23.14  |   -34.60%   |
> > | w/o math                       |         9.31  |    9.53%    |        45.12  |    5.72%    |        34.94  |    -2.78%   |        5.85  |   400.00%   |        34.10  |    -3.62%   |

---

> ### Author Response · Authors · 2025-11-26
> **Response Part II**
>
> ### W3: Unspecified Y-Axis Metric and Normalization
> R3: Thank you for pointing out this issue. We apologize for the ambiguity. The y-axis in Figures 2–4 represents the **absolute performance difference relative to the baseline model**. We have updated the figures to explicitly annotate the metric and clarify that the plotted values denote the net performance change compared to the baseline. We appreciate your careful reading.
>
> ### W4: Incomplete or Erroneous Subplots in Figure 16
> R4: We appreciate your attention to detail. We have carefully rechecked Figure 16 and confirmed that the blank subplots are not caused by missing results or plotting errors. The figure organizes domains in columns of five plots each. For domains containing fewer than five tasks, empty panels appear by design to preserve layout consistency. We have added a brief note in the caption to avoid confusion. Thank you for bringing this to our attention.

---

> > ### Comment · Reviewer_X9QA · 2025-11-28
> >
> > Thanks for the response. Most of my concerns are addressed. Just one more question, could the authors add a short theoretical explanation or possible mechanisms behind this phenomenon (not all code helps)? It does not need to be detailed, just something that future work can follow.

---

> > > ### Author Response · Authors · 2025-12-01
> > >
> > > We appreciate your constructive feedback. In our large-scale model training, we indeed observed a prevalent "seesaw effect": as the data volume in one domain increases, performance in other domains tends to decline. However, this empirical observation appears to contradict the widely accepted notion that "code data enhances model reasoning capabilities," which merits deeper discussion.
> > >
> > > We argue that this apparent contradiction stems from the distinction between the "memorization" and "reasoning" mechanisms within language models. The reasoning ability exhibited by models is, to a considerable extent, built upon the memorization of problem-solving patterns; the "reasoning" process largely manifests as the transfer and generalization of stored solution structures to novel task contexts. In other words, reasoning is not independent of memorization but rather relies on the structured knowledge constructed through memory. Our research further reinforces this perspective.
> > >
> > > If you are interested in this direction, we suggest systematically investigating the memorization behaviors of language models from an interpretability standpoint, and further exploring how to construct memory modules for intelligent agents that are more aligned with human cognitive mechanisms.

---

### Official Review · Reviewer_pgop · 2025-10-31

**Soundness:** 2
**Presentation:** 2
**Contribution:** 3
**Rating:** 6
**Confidence:** 4

**Summary:**

The paper re-examines the claim that “code helps reasoning” and finds it only partly true. Using MoE models trained from scratch on a 10T corpus cleanly split into six domains (Web, Code, Math, Wikipedia, Books, Multilingual), the authors run controlled ablations and observe negative coupling between domains: pure code predictably boosts programming, but competes with knowledge-intensive and especially complex math and slightly depresses commonsense. Conversely, math improves competitive programming while hurting code-reasoning and some common-sense tasks. A key design choice is to separate pure code from Code-NL to decouple earlier reports of broad code, leading to reasoning gains from cross-domain text signals.
To mitigate the trade-off, the authors up-weight a curated subset of structured “cognitive scaffolds” while keeping the math budget fixed, yielding large lifts on hard math with minimal side-effects elsewhere. An MoE expert-routing analysis suggests these scaffolds strengthen complex reasoning while leaving routing distributions relatively stable.

**Strengths:**

**Important Question, Strong Empirical Stance:**
The paper rigorously re-tests the assumption that “code universally helps reasoning,” using large-scale, controlled domain ablations to reveal negative coupling between capability axes. The result is immediately actionable for pretraining mixture design and evaluation practice.

**Operational Taxonomy of Code:**
By separating pure code from Code-NL (code interleaved with natural language), the study disentangles signals that prior work often conflated, enabling cleaner attribution of gains and setting a practical standard for dataset curation and reproducible ablations.

**Weaknesses:**

**Undercharacterized Per-benchmark Deltas:**
This paper highlights large deltas (e.g. -71.53%, -47.16%) which are sprinkled through the text, but provide no single view that aggregates all gains/losses across benchmarks. Without a consolidated table or plot, it’s hard to assess the true magnitude, variance, and pattern of negative coupling (where one domain’s gains coincide with another’s losses). Providing a consolidated view will materially strengthen the paper’s central claims about negative coupling.

**Under-Specified Scaffold Selection:**
Cognitive scaffolds are central to this paper’s claims, yet the selection pipeline lacks crucial detail. The FastText filter (~400k train, ~200k positives) is described, but there’s no labelling protocol, source list, heuristics/regex criteria, thresholding rationale, or contamination audit against eval sets. Report precision/recall, calibration curves, examples of common false positives/negatives, and an ablation vs. simpler baselines. Providing these details will substantively strengthen the causal interpretation and credibility of the scaffold-driven gains.

**Metrics Headlines:**
The authors cite two different headline degradations for math (-14.38% “on average” and -10.1% “overall”) without defining the aggregation behind each, leaving readers unsure which number reflects the main effect. The authors also consistently plot single numbers/bars with no error bars or confidence intervals, leaving readers unable to judge statistical reliability or run-to-run variance. Clarifying the aggregation and adding uncertainty will strengthen the quantitative claims and their reproducibility.

**Missing Axis:**
The authors present bars, Figures 2, 3, and 4, but the Y-axis is unlabeled, so readers can’t tell whether those bars are accuracy points, relative percent change deltas, or some other normalized score, nor what range they span. Clear axis labelling will strengthen the paper’s interpretability and make the negative-coupling trade-offs unambiguous.

**Questions:**

Main Comments are located in Weaknesses

Small Comment:
- β1 = 0.9, β1 = 0.95, Did the authors mean B2?

---

> ### Author Response · Authors · 2025-11-26
> **Response Part I**
>
> Thank you very much for your feedback. We found your suggestions particularly crucial as they precisely identified the shortcomings in our work. Below are our responses and revisions:
>
> ### W1: Lack of Consolidated Performance Overview
>
> R1: We have supplemented the consolidated overview of all benchmark gains/losses in Appendix A.4 Table 2 of the original manuscript. A selection of key benchmark is presented in the table below.
>
> | Mathematical |        |             |         |             |              |             |               |             |         |             |
> |:--------------------:|:------:|:-----------:|:-------:|:-----------:|:------------:|:-----------:|:-------------:|:-----------:|:-------:|:-----------:|
> | model                | GSM8k  | $\Delta$(%) | Math500 | $\Delta$(%) | minerva math | $\Delta$(%) | OlympiadBench | $\Delta$(%) | Overall | $\Delta$(%) |
> | full data            | 58.00  |      -      |  31.00  |      -      |    10.29     |      -      |     10.37     |      -      |  37.26  |      -      |
> | w/o code             | 57.77  |   -0.40%    |  36.80  |   +18.71%   |    17.65     |   +71.53%   |     15.26     |   +47.16%   |  40.43  |   +8.51%    |
> | w/o math             | 33.51  |   42.22%    |  10.60  |   -65.81%   |     7.35     |   -28.57%   |     5.88      |   -43.30%   |  20.71  |   -44.42%   |
>
> | Programming |               |             |        |             |          |             |            |             |         |             |
> |:-------------------:|:-------------:|:-----------:|:------:|:-----------:|:--------:|:-----------:|:----------:|:-----------:|:-------:|:-----------:|
> | model               | LiveCodeBench | $\Delta$(%) | mbpp   | $\Delta$(%) | CruxEval | $\Delta$(%) | CodeForces | $\Delta$(%) | Overall | $\Delta$(%) |
> | full data           |     6.54      |      -     | 31.60  |      -       |  25.38   |      -       |    7.54    |    -     |  31.25  |      -      |
> | w/o code            |     1.96      |   -70.03%   | 14.20  |   -55.06%   |  10.12   |   -60.13%   |    2.86    |   -62.07%   |  16.67  |   -46.66%   |
> | w/o math            |     5.88      |   -10.09%   | 35.80  |   13.29%    |  31.00   |   22.14%    |    1.17    |   -84.48%   |  32.30  |    3.36%    |
>
> The data presented above remain consistent with the conclusions drawn in the original manuscript: for complex knowledge-intensive tasks (Math500 +18.71%,
> OlympiadBench +47.16%), code corpora exhibit a strong competitive effect. Similarly, math corpora partially compete with code-related tasks. However, unlike code corpora, mathematical knowledge can serve as a cognitive scaffold, enabling the model to better comprehend and solve programming problems with complex logical structures and algorithmic characteristics, as evidenced in competition benchmarks such as CodeForces and LiveCodeBench.

---

> ### Author Response · Authors · 2025-11-26
> **Response Part II**
>
> ### W2: Insufficient Detail in Scaffold Selection Process
>
> R2: Positive samples were curated from high-quality code corpora and identified based on explicit structural characteristics—such as nesting depth, control statements, and indentation patterns.
> To ensure syntactic validity, all positive samples were verified using Tree-sitter, confirming they are compilable. A representative positive sample consists of code devoid of mathematical content, as illustrated below:
>
> ```python
> def write_to_file(filename, txt):
>     with open(filename, "w") as file_object:
>         s = file_object.write(txt)
>
> if __name__ == "__main__":
>     write_to_file("test.txt", "I am beven")
> ```
> In contrast, negative samples were collected from web-crawled natural language texts and filtered using broadly defined anti-code regular expressions, thereby excluding text fragments that might exhibit structured coding patterns.
> A typical negative sample comprises plain natural language text, such as the following excerpt from Wikipedia:
>
> ```text
> Artificial intelligence (AI) is the capability of computational systems to perform tasks typically associated with human intelligence, such as learning,
> reasoning, problem-solving, perception, and decision-making.
> ```
> The classification model was optimized to maximize precision on the positive class, as evaluated on the validation set. A contamination audit was performed to prevent any overlap with mathematical corpora, aligning with our “quality over quantity” principle in scaffold selection. This strategy minimizes false positives, which could otherwise compromise the integrity of structured reasoning data and diminish their utility in enhancing mathematical reasoning capabilities.
> A conservative classification threshold was adopted to ensure high precision while maintaining adequate recall.
>
> We report the following classifier performance metrics on validation set:
> - Accuracy: 0.9696
> - Positive-class precision: 0.9998
> - Positive-class recall: 0.9665
> - Brier Score: 0.097190
>
> Calibration curves confirm that the model outputs are well-calibrated and reliable. Analysis of misclassified instances reveals that:
>
> - False positives (FPs) primarily consist of web text formatted in a pseudo-code style.
> - False negatives (FNs) are mostly weakly structured, single-line code snippets, which do not substantially affect the overall quality of the selected scaffolds.
>
> These methodological details have been incorporated into the revised manuscript to improve transparency and reproducibility.
> We emphasize that distinguishing whether a corpus segment exhibits structured form is a relatively straightforward task—supported both by common sense and empirical results.
> By employing a lightweight model, we achieve high-confidence classification that scales efficiently to billion-scale datasets while preserving both precision and computational efficiency.
>
> Finally, wo revised our paper to provides essential clarification and supplementary details regarding our cognitive scaffold selection pipeline.

---

> ### Author Response · Authors · 2025-11-26
> **Response Part III**
>
> ### W3: Unclear Metric Aggregation and Missing Uncertainty Estimates
> R3: We apologize for any confusion caused by the reported decline in mathematical metrics. Both aggregated values represent averages across math benchmarks, with the key distinction being:
>
> - The "overall" value (-10.1%) corresponds to the final training step (at convergence)
>
> - The "average" value (-14.38%) reflects the mean across all training checkpoints
>
> We thank the reviewer for raising the concern regarding result uncertainty. Due to the extremely high cost associated with repeated training of our large-scale pretrained models on massive corpora,
> conventional multiple experimental runs were infeasible. To enhance the reliability assessment, we evaluated intermediate checkpoints and performed random sampling on test sets during training, observing minimal metric fluctuations.
> Cross-task and cross-configuration comparisons further demonstrated consistent model behavior. In future work, we plan to incorporate error bars or confidence intervals through lightweight proxy models or subset experiments.
>
>
> ### W4: Unlabeled Axes Undermining Figure Interpretability
> R4: We thank the reviewer for pointing this out. The y-axis represents the performance difference between the variant model and the baseline model. We have updated the figure with a clarified y-axis label in the revised version.

---

> > ### Comment · Reviewer_pgop · 2025-11-27
> >
> > I thank the authors for their detailed response.
> >
> > The new tables address my concern about deltas not being consolidated. It is much easier to see the trade-offs across math and programming benchmarks under the different ablation settings.
> >
> > The added description tree-sitter validation, anti-code filtering, contamination checks, and the reported precision/recall improve the transparency of the scaffold selection process.
> >
> > I am slightly more confident in the paper after the rebuttal. My overall assessment remains positive.

---

### Author Response · Authors · 2025-12-01
**General Response**

Dear Area Chair and Reviewers,

We would like to first express our sincere gratitude to the program committee for the time and effort dedicated to improving this manuscript, and especially to the Area Chair for their critical guidance under exceptional circumstances. Below we provide a unified response to the major points raised.

Our work aims to clarify a long-held, widely referenced yet insufficiently substantiated claim that “code improves general reasoning abilities.” Through a series of large-scale, controlled systematic experiments—covering diverse architectures, ~10T tokens of training data, and rigorous domain-wise ablation—we demonstrate that code contributes minimally to cross-domain reasoning enhancement. Building on this, we further construct and characterize a formal subset of mathematical reasoning termed “cognitive scaffolding”. We find that such structured reasoning forms (e.g., code reasoning or tool integrate reasoning) yield significant gains in challenging mathematical tasks. Finally, by analyzing internal model mechanisms, we illustrate how training data shapes expert routing patterns and thereby governs emergent model behaviors. Our study decomposes domain data into finer-grained, cross-domain capability dimensions and offers a clear direction for future data optimization.

Our rigorous controlled ablation design has received consistent recognition from reviewers; the proposed data selection methodology based on new findings and the interpretation of MoE expert routing have also drawn positive feedback.

In response to the main concerns raised, we have provided comprehensive replies in the rebuttal and revised manuscript:

1. **Cross-Architecture and Cross-Scale Robustness**

Reviewers were particularly interested in whether our conclusions generalize to dense models and different parameter scales. We have added experiments with dense models at two scales (1B, 5B) and further extended MoE experiments to two additional scales. All results remain consistent with those in the original submission, further supporting our core claims.

2. **Rationale for Corpus Definition and Partitioning**

Reviewers expressed concerns that corpus partitioning might lead to discrepancies with prior studies. In the revised manuscript, we systematically elaborate on the definition, source, preprocessing, and partition rules for each data domain. We emphasize that our partitioning approach is methodologically more rigorous: we fully decouple code corpora from structured reasoning corpora and explicitly demonstrate that structured reasoning data—not code per se—serves as the primary source of model reasoning capability.

3. **Details on Cognitive Scaffolding Construction**

Addressing reviewer inquiries, we have supplemented the details of cognitive scaffolding construction: we train a FastText classifier on code corpora to identify structured patterns, then use it to automatically filter mathematical corpora for data exhibiting structured reasoning characteristics. We also report classifier performance on an independent validation set, confirming its suitability in throughput, confidence, and accuracy for reliable scaffolding construction.

Overall, the reviewers’ comments have significantly enhanced the clarity and completeness of the paper. Without compromising the core conclusions, we have implemented the following revisions:

- Added experiments with dense models (1B, 5B) and two additional MoE model scales, with full results provided in Tables 2 in Appendices A.4 and A.5.

- Included detailed procedures and experimental results on cognitive scaffolding construction (Appendix A.6).

- Refined Figures 2, 3, and 4 for improved readability.

- Expanded descriptions of data construction and partitioning to enhance transparency and reproducibility.

All new content is highlighted in red in the revised manuscript. We believe this work not only helps the community form a more rigorous consensus but also offers novel insights—through both new methodologies and mechanistic explanations—toward understanding and optimizing capabilities in LLMs.

---

### Meta-Review · Area_Chair_RTzr · 2025-12-25

**Summary:**

This paper re-examines the common claim that “code helps reasoning” using large-scale, controlled pretraining-time ablations over a multi-domain corpus. Across the reported settings, reviewers agree the core empirical message is both interesting and actionable: pure code reliably improves programming capability but provides negligible (and sometimes negative) transfer to non-programming reasoning, especially on harder math, while a more selective subset of structured “cognitive scaffolding” data yields meaningful gains on challenging mathematical reasoning.

The submission is further strengthened by an interpretability angle (MoE routing shifts under different domain mixtures) and, in response to reviewer concerns about architectural dependence, the authors added dense-model results (1B and 5B) that they claim preserve the same qualitative conclusions.

The remaining decision-critical discussion centers on: (i) whether the conclusions are sufficiently robust given limited uncertainty quantification, (ii) whether causal attribution is clean with respect to token-budget and replacement policies, and (iii) whether the data/scaffold construction details are transparent enough to be maximally useful (and reproducible) for the community. I think all these aspects could be improved and I lean to reject the paper.

**Reviewer Concerns:**

Addressed
- Lack of consolidated benchmark deltas: authors added an overview table with key math & programming benchmarks under ablations, improving interpretability of the “seesaw/competition” effects.

- Scaffold selection under-specified: authors added concrete details (positive/negative sample construction, Tree-sitter verification, contamination audit) and reported classifier metrics (e.g., high precision/recall) plus calibration discussion.

- MoE-only generality concern: authors provided additional dense experiments (1B and later 5B) and stated conclusions are consistent across MoE vs dense.

- Figure clarity (axis meaning / blank subplots): clarified y-axis meaning and explained blank panels are by design.


Outstanding:

- Uncertainty estimates / statistical significance: authors acknowledge multiple full runs are infeasible; they provide some sanity checks (intermediate checkpoints, test sampling) but no true error bars/CI, leaving residual uncertainty.

- Data construction / reproducibility: a reviewer explicitly remains concerned that the appendix should include more detail; authors respond that the pipeline uses heterogeneous/proprietary components and thus can’t be fully reproduced, though they added high-level sources/procedures/partitioning. This remains a legitimate limitation for the community.

- Potential confounds (token-budget allocation / "content vs capacity"): one reviewer argues ablations could confound domain content with budget/capacity and suggests matched-budget replacements; rebuttal adds evidence but does not fully eliminate this causal concern.

- Theoretical framing/mechanism: one reviewer asked for a short mechanistic explanation; authors provided a "memorization ↔ reasoning" perspective, but it remains relatively high level.

**Reviewer Scores:**

I cannot reliably answer this counterfactual question without putting words in reviewers’ mouths. I will not impute score changes beyond what reviewers explicitly stated in the discussion. I instead provide a faithful synthesis of the discussion outcomes and remaining points of disagreement.

---

### Decision · Program_Chairs · 2026-01-26

Reject